# Automated hippocampal unfolding for morphometry and subfield segmentation with HippUnfold

Jordan DeKraker[1,2]*, Roy AM Haast[1], Mohamed D Yousif[1], Bradley Karat[1], Jonathan C Lau[1,3,4], Stefan Köhler[2,5], Ali R Khan[1,2,4,6]*

[1]Robarts Research Institute, Schulich School of Medicine and Dentistry, The University of Western Ontario, London, Canada; [2]Western Institute for Neuroscience, The University of Western Ontario, London, Canada; [3]Department of Clinical Neurological Sciences, Division of Neurosurgery, Schulich School of Medicine & Dentistry, The University of Western Ontario, London, Canada; [4]School of Biomedical Engineering, The University of Western Ontario, London, Canada; [5]Department of Psychology, Faculty of Social Science, The University of Western Ontario, London, Canada; [6]Department of Medical Biophysics, Schulich School of Medicine and Dentistry, The University of Western Ontario, London, Canada

**Abstract** Like neocortical structures, the archicortical hippocampus differs in its folding patterns across individuals. Here, we present an automated and robust BIDS-App, HippUnfold, for defining and indexing individual-specific hippocampal folding in MRI, analogous to popular tools used in neocortical reconstruction. Such tailoring is critical for inter-individual alignment, with topology serving as the basis for homology. This topological framework enables qualitatively new analyses of morphological and laminar structure in the hippocampus or its subfields. It is critical for refining current neuroimaging analyses at a meso- as well as micro-scale. HippUnfold uses state-of-the-art deep learning combined with previously developed topological constraints to generate uniquely folded surfaces to fit a given subject's hippocampal conformation. It is designed to work with commonly employed sub-millimetric MRI acquisitions, with possible extension to microscopic resolution. In this paper, we describe the power of HippUnfold in feature extraction, and highlight its unique value compared to several extant hippocampal subfield analysis methods.

*For correspondence:
jordan.dekraker@mail.mcgill.ca (JDeK);
ali.khan@uwo.ca (ARK)

**Competing interest:** The authors declare that no competing interests exist.

## Editor's evaluation

This study presents a useful automated package called 'HippUnfold' in form of a BIDS App. The approach is solid and validated by comparing it against other methods in the field and has the potential to be used by a wide audience.

## Introduction

Most neurological or psychiatric diseases with widespread effects on the brain show strong and early impact on the hippocampus (e.g. *Thom, 2014*). This highly plastic grey matter (GM) structure is also critical in the fast formation of episodic and spatial memories (e.g. *O'Keefe and Nadel, 1978*). Examination of this structure with non-invasive neuroimaging, such as MRI, provides great promise for furthering our understanding, diagnosis, and subtyping of these diseases and cognitive processes in the hippocampus and its component subfields (*Dill et al., 2015*).

In current neuroimaging analyses the hippocampus is typically modelled as a subcortical volume, but it is actually made up of a folded archicortical mantle, or 'ribbon' (*Duvernoy, 1998*). Representing the hippocampus as such can be leveraged to enable qualitatively new analyses, such as registration, despite inter-individual differences in gyrification or folding structure, through topological alignment. In previous work, this was shown to account for much inter-individual variability in MRI-based manual subfield segmentations (*DeKraker et al., 2018*). Additionally, representation as a ribbon allows the hippocampus to be factorized into surface area and thickness, which can be further subdivided for laminar analyses. These methods are thus critical in advancing MRI research from the macroscopic scale to the subfield, cortical column, and laminar scales. Similar approaches have already yielded advances in neocortical analysis methods (*Van Essen et al., 2000*; *Waehnert et al., 2014*).

Denoting the hippocampal archicortical ribbon is challenging because it is thin (0.5–2 mm), its folding pattern varies considerably between individuals (*Chang et al., 2018*; *Ding and Van Hoesen, 2015*), and this folding may even continue to change from early development through adulthood (*Cai et al., 2019*). We present here a set of tools to overcome these challenges using a highly sensitive and generalizable 'U-Net' deep learning architecture (*Isensee et al., 2021*), combined with previous work that enforces topological constraints on hippocampal tissue (*DeKraker et al., 2018*).

In previous work (*DeKraker et al., 2018*), we developed a method to computationally unfold the hippocampus along its geodesic anterior-posterior (AP) and proximal-distal (PD, i.e. proximal to the neocortex, with the dentate gyrus [DG] being most distal) axes. We demonstrated for the first time several qualitative properties using in vivo MRI, such as the contiguity of all subfields along the curvature of the hippocampal head (anterior) and tail (posterior), previously described only in histology. This pioneering work relied heavily on detailed manual tissue segmentations including the high-myelinated stratum radiatum, lacunosum, and moleculare (SRLM), a commonly used landmark that separates hippocampal folds along the inward 'curl' of the hippocampus. In this work we also considered curvature and digitations along the AP axis of the hippocampus, most prominently occurring in the hippocampal head (*Duvernoy, 1998*; *Chang et al., 2018*; *Ding and Van Hoesen, 2015*; *DeKraker et al., 2018*). Each of these features are highly variable between individuals, making them difficult to capture using automated volumetric atlas-based methods and time-consuming to detect manually.

The current work automates the detailed tissue segmentation required for hippocampal unfolding using a state-of-the-art 'U-Net' deep convolutional neural network (*Isensee et al., 2021*). In particular, we aimed to capture morphological variability between hippocampi at the level of digitations or gyrifications which are not typically considered using existing automated methods which employ either a single atlas or multi-atlas fusion (e.g. *Yushkevich et al., 2015b*; *Chakravarty et al., 2013*; *Pipitone et al., 2014*). U-Net architectures have been shown to be generalizable and sensitive to anatomical variations in many medical image processing tasks (*Du et al., 2020*), making them ideal to overcome this challenge.

Estimating hippocampal subfield boundaries in MRI is challenging since their histological hallmarks are not directly available in MRI due to lower spatial resolution and lack of appropriate contrasts, which is an ongoing hurdle in neuroimaging (*Wisse et al., 2017b*; *Yushkevich et al., 2015a*). However, post-mortem studies show that the subfields are topologically constrained according to their differentiation from a common flat cortical mantle (*Duvernoy, 1998*). Thus, a folded representation of hippocampal tissue provides a powerful intermediate between a raw MRI and subfield labels (*DeKraker et al., 2021*), analogous to the reconstruction of a 3D neocortical surface. This surface can then be parcellated into subregions without topological breaks (*Van Essen et al., 2000*), overcoming many limitations of current subfield segmentation methods (*Yushkevich et al., 2015a*). Here, we apply surface-based subfield boundary definitions obtained via manual segmentation of BigBrain 3D histology (*Amunts et al., 2013*) which was additionally supported by a data-driven parcellation (*DeKraker et al., 2020*). We additionally demonstrate how labels used in the popular Freesurfer (FS7) (*Iglesias et al., 2015*) and Automatic Segmentation of Hippocampal Subfields (ASHS) (*Yushkevich et al., 2015b*) software packages can be applied under our topologically constrained framework.

Altogether, we combine novel U-Net tissue classification, previously developed hippocampal unfolding (*DeKraker et al., 2018*), and topologically constrained subfield labelling (*DeKraker et al., 2020*) together into a single pipeline which we refer to as 'HippUnfold' hereinafter. We designed this pipeline to employ FAIR principles (findability, accessibility, interoperability, reusability) with support across a wide range of use-cases centred around sub-millimetric MRI.

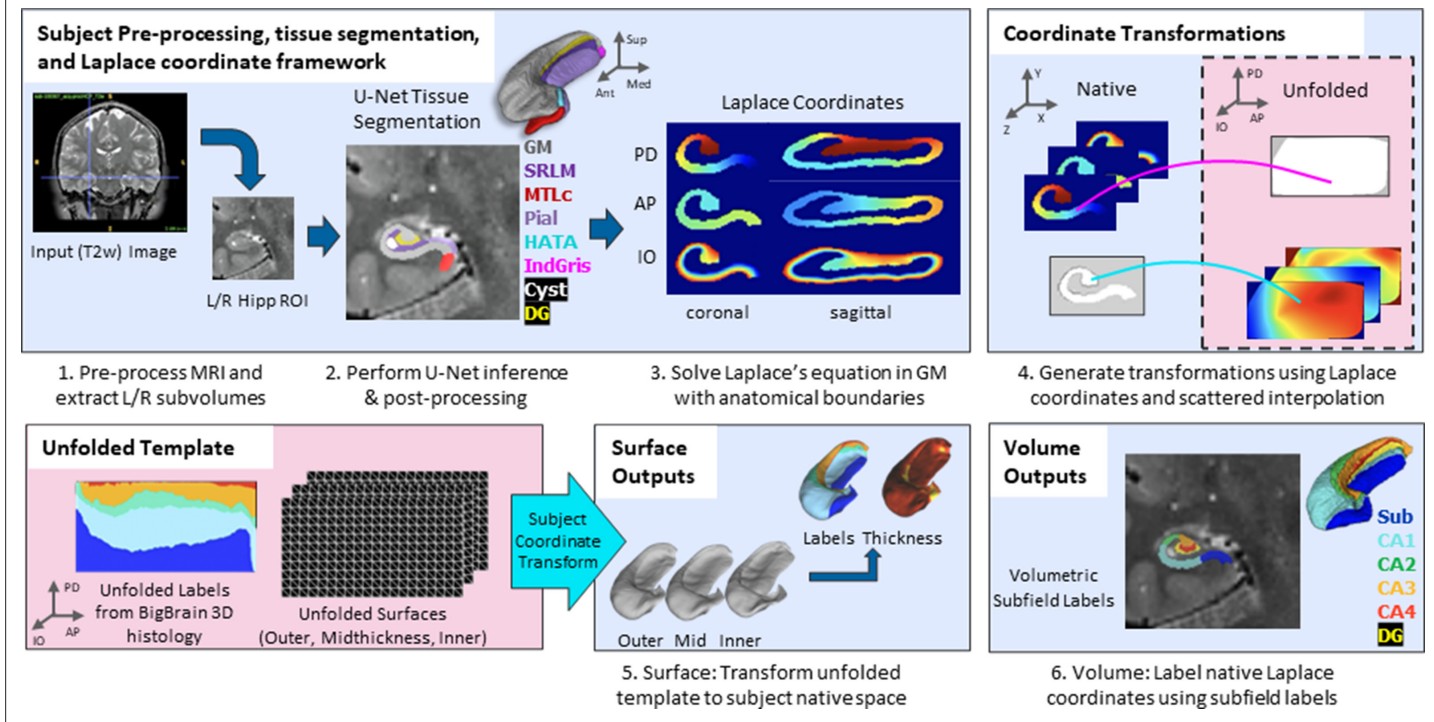

**Figure 1.** Overview of HippUnfold pipeline. First, input MRI images are preprocessed and cropped around the left and right hippocampi. Second, a U-Net neural network architecture (nnUNet; *Isensee et al., 2021*) is used to segment hippocampal grey matter (GM), the high-myelinated stratum radiatum, lacunosum, and moleculare (SRLM), and structures surrounding the hippocampus. Segmentations are post-processed via template shape injection. Third, Laplace's equation is solved across the anterior-posterior (AP), proximal-distal (PD), and inner-outer (IO) extent of hippocampal GM, making up a geodesic coordinate framework. Fourth, scattered interpolants are used to determine equivalent coordinates between native Cartesian space and unfolded space. Fifth, unfolded surfaces with template subfield labels (*DeKraker et al., 2020*) are transformed to subjects' native folded hippocampal configurations. Morphological features (e.g. thickness) are extracted using Connectome Workbench (*Glasser et al., 2013*) on these folded native space surfaces. Sixth, volumetric subfields are generated by filling the voxels between inner and outer surfaces with the corresponding subfield labels. Additional details on this pipeline can be found in the Materials and methods.

The online version of this article includes the following figure supplement(s) for figure 1:

**Figure supplement 1.** Diagram of the nnU-net architecture used for HippUnfold.

## Results

### HippUnfold aligns and visualizes data on folded or unfolded surfaces

HippUnfold is presented here as a fully automated pipeline with outputs including hippocampal tissue and subfield segmentations, geodesic Laplace coordinates spanning over hippocampal GM voxels, and inner, midthickness and outer hippocampal surfaces. These surfaces have corresponding vertices, providing an implicit topological registration between individuals.

The overall pipeline for HippUnfold is illustrated briefly in *Figure 1*. A comprehensive breakdown of each step is provided in the Materials and methods.

In addition to subfield segmentation, HippUnfold extracts morphological features and can be used to sample quantitative MRI data along a midthickness surface to minimize partial voluming with surrounding structures (see Materials and methods section 'HippUnfold detailed pipeline' for details). This is visualized across n=148 test subjects on an unfolded surface and group-averaged folded surface in *Figure 2*. Note that the group averaging takes place on a surface and so does not break individual subjects' topologies. Quantitative MRI features examined here include T1w/T2w ratio as a proxy measure for intracortical myelin (*Ganzetti et al., 2014*), mean diffusivity, and fractional anisotropy (*Hernández et al., 2013*; *Sotiropoulos et al., 2016*).

Clear differences in morphological and quantitative MRI features can be seen across the hippocampus, particularly across subfields as defined here from a histologically derived unfolded reference atlas (3D BigBrain) (*DeKraker et al., 2020*). This highlights the advantages of the present method.

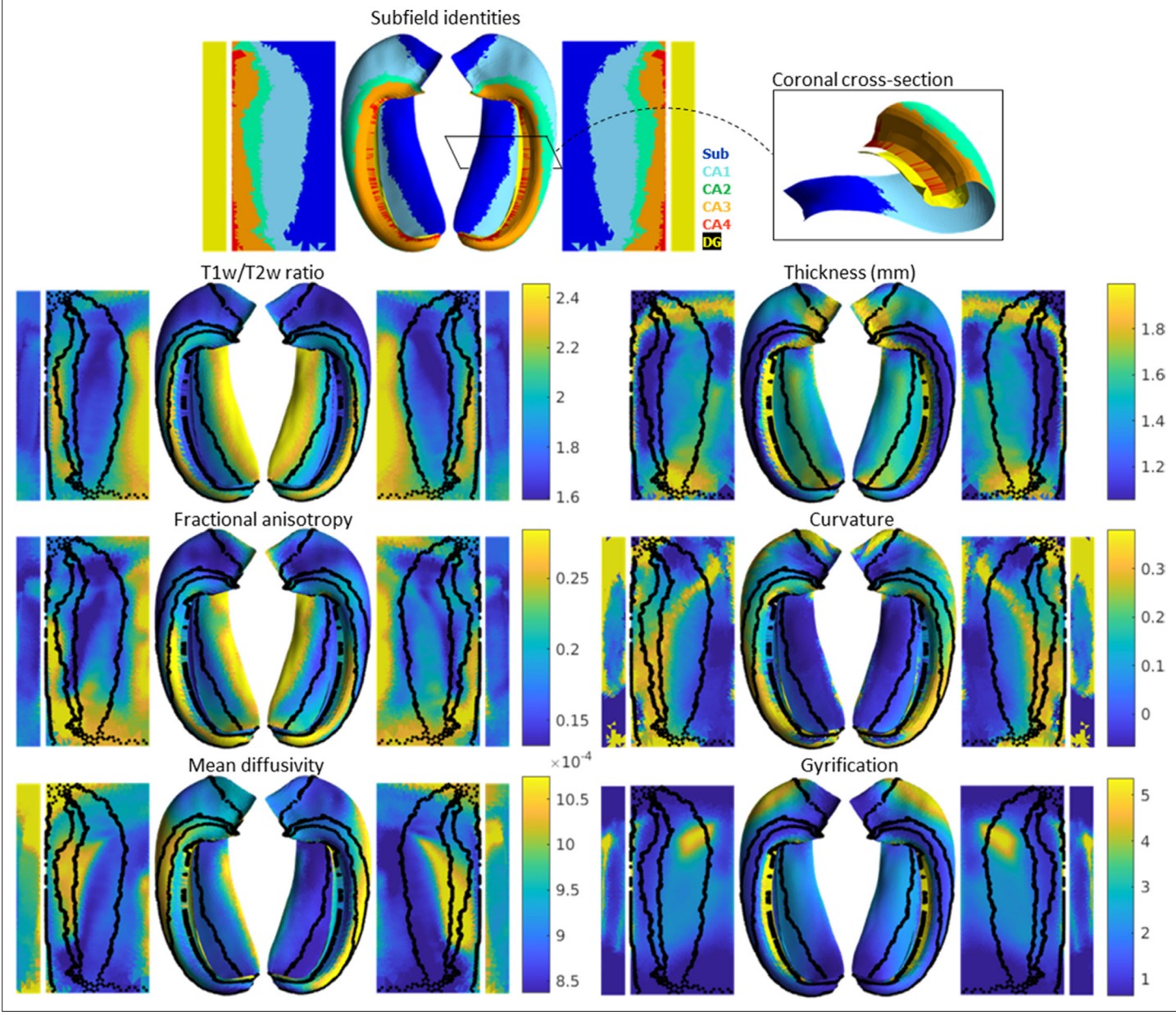

**Figure 2.** Average hippocampal folded and unfolded surfaces showing subfields, morphometric, and quantitative MRI measures from the Human Connectome Project-Young Adult (HCP-YA) test dataset (see *Table 1* of Materials and methods). The same topologically defined subfields were applied in unfolded space to all subjects (top), which are also overlaid on quantitative MRI plots (black lines). The dentate gyrus (DG) is represented as a distinct surface, reflecting its unique topology, and is mostly occluded in native space. Thickness was not measured across the DG surface. Note that many morphological and quantitative MRI measures show clear distinctions across subfield boundaries.

The online version of this article includes the following figure supplement(s) for figure 2:

**Figure supplement 1.** Examination of distortions (or difference in vertex spacing) between an average folded and unfolded space.

These folded and unfolded representations of hippocampal characteristics are broadly in line with previous work examining differences in such morphological and quantitative MRI features across hippocampal subfields or along the hippocampal AP extent (e.g. *Vos de Wael et al., 2018*; *Crombe et al., 2018*). However, in previous work these features differed between predefined subfields on average, but did not necessarily follow subfield contours as seen here. Some advantages of the current pipeline that likely contribute to this clarity include (i) the detail of the hippocampal GM segmentation, (ii) sampling along a midthickness surface to minimize partial voluming with surrounding structures, and

(iii) the fact that subjects are topologically aligned across digitations or gyri, leading to less blurring of features after group-averaging.

## Extant methods do not respect the topological continuity of hippocampal subfields

Several automatic methods for labelling hippocampal subfields in MRI exist, of which FS7 (*Iglesias et al., 2015*) and ASHS (*Yushkevich et al., 2015b*) are among the most widely adopted. These methods rely on volumetric registrations between a target hippocampus and a reference or atlas. Specifically, ASHS makes use of multi-atlas registration, wherein multiple gold standard manual hippocampal subfield segmentations are registered to a target sample. Typically the multi-atlas consists of roughly a dozen samples which are then fused together to generate a reliable yet oftentimes smooth or simplified final product. FS uses a combination of voxel-wise classification and, bijectively, volumetric registration between a target hippocampus and a probabilistic reference atlas, which is generated via combined in vivo MRI and 9.4T ex vivo hippocampal subfield segmentations (*Iglesias et al., 2015*). When hippocampi take on different folding configurations, such registrations can become ill-posed. HippUnfold overcomes these limitations in two ways: with extensive training (in this case n=590), U-Net can capture detailed inter-individual differences in folding and, secondly, our unfolding technique ensures that subfield labelling is topologically constrained (*DeKraker et al., 2021*).

We made use of 100 randomly chosen subjects from the Human Connectome Project-Aging (HCP-A) dataset to compare the approach with the FS7 hippocampal subfields pipeline and ASHS using a recent manual subfield multi-atlas (*Berron et al., 2017*). *Figure 3A* shows a side-by-side comparison of HippUnfold, ASHS, and FS7 to one representative 81-year-old female. *Figure 3B* shows Bland-Altmann plots comparing subfields CA1, CA3, and subiculum volume across the three methods in all 100 subjects, as well as their correlation with subjects' ages. Quantitative comparison between methods shows an age-related decline in subfield volumes for all methods, with a relative sparing of CA3. Thus, HippUnfold replicates the widely observed phenomenon of age-related decline, with similar effect sizes to FS and ASHS (*Figure 3—figure supplement 1*). A similar pattern can be seen across the other subfield volumes and in total hippocampal volume. Bland-Altman plots show major differences in hippocampal subfield sizes between methods, which most likely results from inclusion of the hippocampal tail in HippUnfold.

Within the HCP-YA test set, we compared subfield segmentations from ASHS and FS7 to those generated via HippUnfold in unfolded space, which is shown in *Figure 4* in one representative subject. We then generated an unfolded subfield atlas using the maximum probability labels from all ASHS and FS7 subjects, which can be used in place of the default HippUnfold atlas generated via 3D BigBrain histology (*DeKraker et al., 2020*). For comparison, we additionally show native space HippUnfold results obtained when using these alternative unfolded atlases.

Both ASHS and FS showed subfield discontinuities in unfolded space in at least some subjects, and FS even showed discontinuities in the group-averaged unfolded subfields. That is, some pieces of a given label were separated from the rest of that label. ASHS does not include an SRLM label and the SRLM produced by FS was not consistently aligned with that used in unfolding. Thus, subfields sometimes erroneously crossed the SRLM, breaking topology and explaining why discontinuities were sometimes observed in unfolded space. Ordering of labels was also not consistent in ASHS and FS. For example, sometimes CA1 would border not only CA2 but also CA3, CA4, and/or DG. Additionally, neither ASHS nor FS extends all subfields to the full anterior and posterior extent of the hippocampus. Instead, both methods simplify most of the anterior hippocampus as being CA1 and opt not to label subfields in the posterior hippocampus at all. These qualities are not in line with the anatomical ground truth shown in both classic and contemporary ex vivo histological studies (*Duvernoy, 1998*; *Ding and Van Hoesen, 2015*), which were indeed well captured by HippUnfold. FS also over-labelled hippocampal tissue, which can be seen reaching laterally into the ventricles in the coronal view. Similar errors have been documented for FS in other recent work (*Wisse et al., 2014*; *Haast et al., 2021*).

## Trained U-Net performance is similar to manual segmentation

From the HCP-YA dataset, a set of 738 (left and right from 369 subjects) gold standard hippocampal tissue (i.e. hippocampal GM and surrounding structures) segmentations were generated according to the manual protocol defined in *DeKraker et al., 2020*. Specifically, this was done by raters JD, MY,

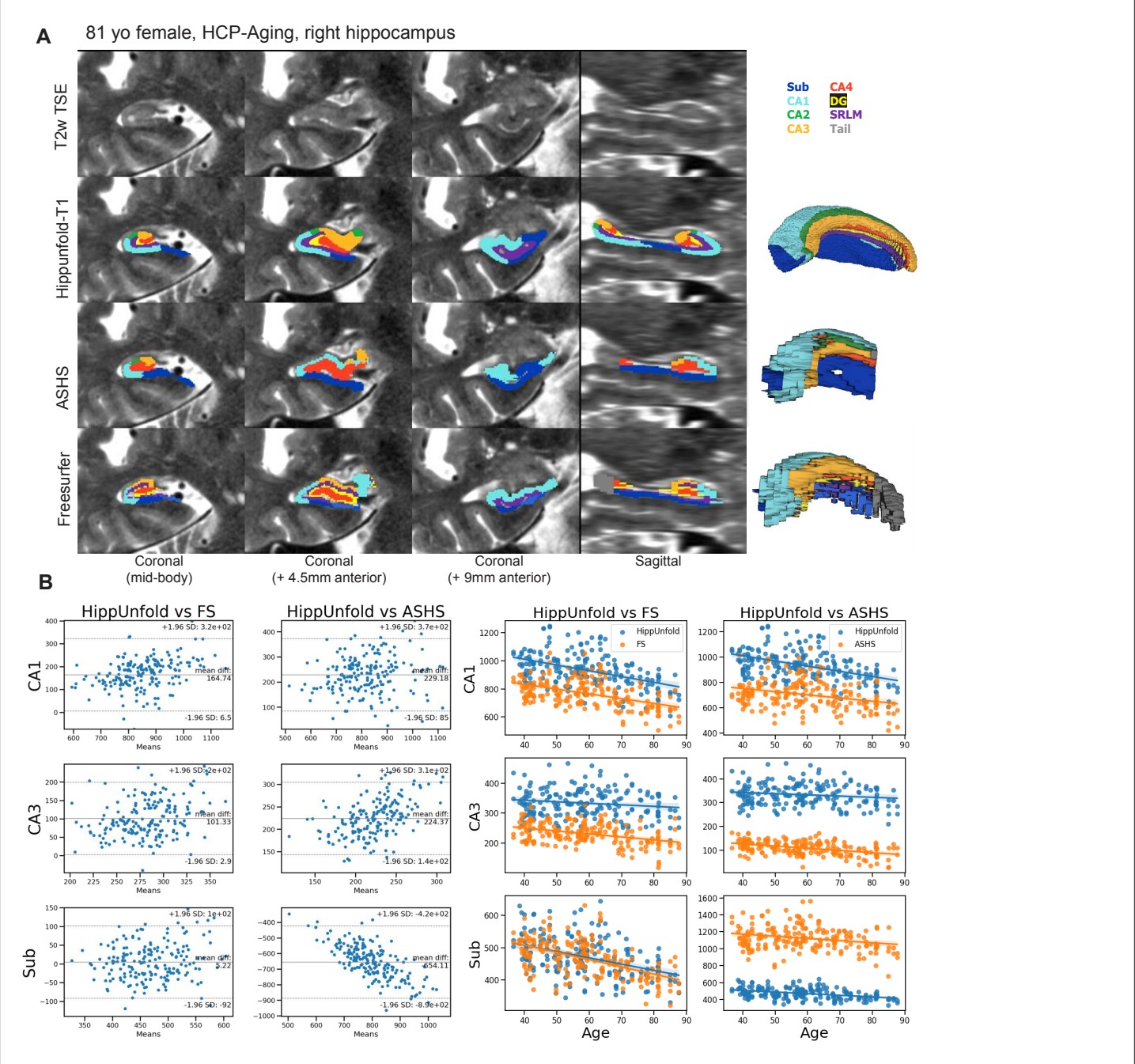

**Figure 3.** Out of sample performance of HippUnfold, Automatic Segmentation of Hippocampal Subfields (ASHS), and Freesurfer (FS7). (**A**) Side-by-side comparison of results obtained from each method from one representative individual from the Human Connectome Project-Aging (HCP-A) datasets, which was not seen during training. (**B**) Quantitative comparison of subfield volumes (left) and age-related volume changes (right) between methods. For a full set of snapshots illustrating the differences between these methods, see *Supplementary file 2*, *Supplementary file 3*.

The online version of this article includes the following figure supplement(s) for figure 3:

**Figure supplement 1.** Additional comparisons of results obtained from Freesurfer (FS7), Automatic Segmentation of Hippocampal Subfields (ASHS), and HippUnfold in 100 Human Connectome Project- Aging (HCP-A) subjects.

and BK using an incremental learning U-Net training regime described in the Materials and methods 'nnUNet training' section. Automated tissue segmentation was performed using nnUNet, a recent and highly generalizable implementation of a U-Net architecture (*Isensee et al., 2021*). This software was trained on 80% (n=590) of the gold standard segmentation data described above, with the remaining

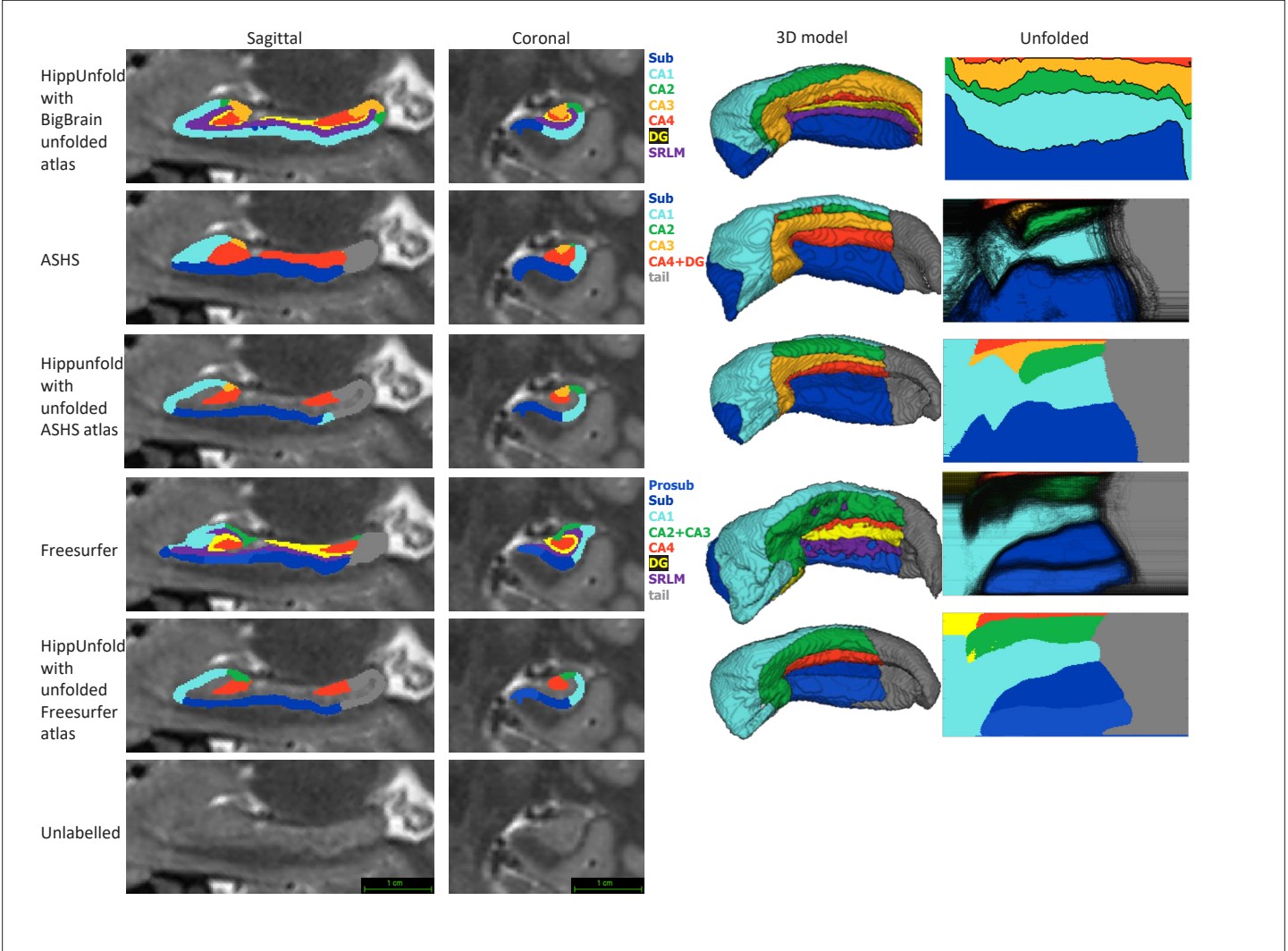

**Figure 4.** Comparison of HippUnfold, Automatic Segmentation of Hippocampal Subfields (ASHS), and Freesurfer (FS7) subfield segmentations in native and unfolded space. Sagittal and coronal slices and 3D models are shown for one representative subject. Note that for HippUnfold hippocampal subfields are the same for all individuals in unfolded space, but for ASHS and FS we mapped all subjects' subfield boundaries which are shown in the black lines in column 4 rows 2 and 4. We then took the maximum probability subfield label from ASHS and FS in unfolded space and used it for HippUnfold subfield segmentation in native space, which is shown in rows 3 and 5.

The online version of this article includes the following figure supplement(s) for figure 4:

**Figure supplement 1.** Comparison of HippUnfold and fully manual subfield segmentations (data from *Thom, 2014*) in native and unfolded space from one representative subject.

20% (n=148) making up a test set. Dice overlap scores on the test set are shown in *Figure 5*. Left and right hippocampi from the same participant were never split across training and testing sets due to their high symmetry. Note that all input images were preprocessed, resampled, and cropped (see *Figure 1* and Materials and methods) prior to training. Within the training set, fivefold cross-validation was performed as implemented in the nnUNet code. Training took place on an NVIDIA T4 Turing GPU over 72 hr. This process was carried out using either T1w or T2w input data with the same training/ testing data split. All default nnUNet data augmentation and hyperparameters were used.

Dice overlap depends heavily on the size of the label in question, being lower for smaller labels. Typically a score of >0.7 is considered good, and many fully manual protocols show dice scores of >0.8 for the larger subfields like CA1 or the subiculum, and 0.6–0.8 for smaller subfields like CA2 or CA3 (see *Yushkevich et al., 2015a*, for overview). Within the HCP-YA test set, performance was similar or better than most fully manual protocols for T1w and T2w data. Performance on T1w images was only

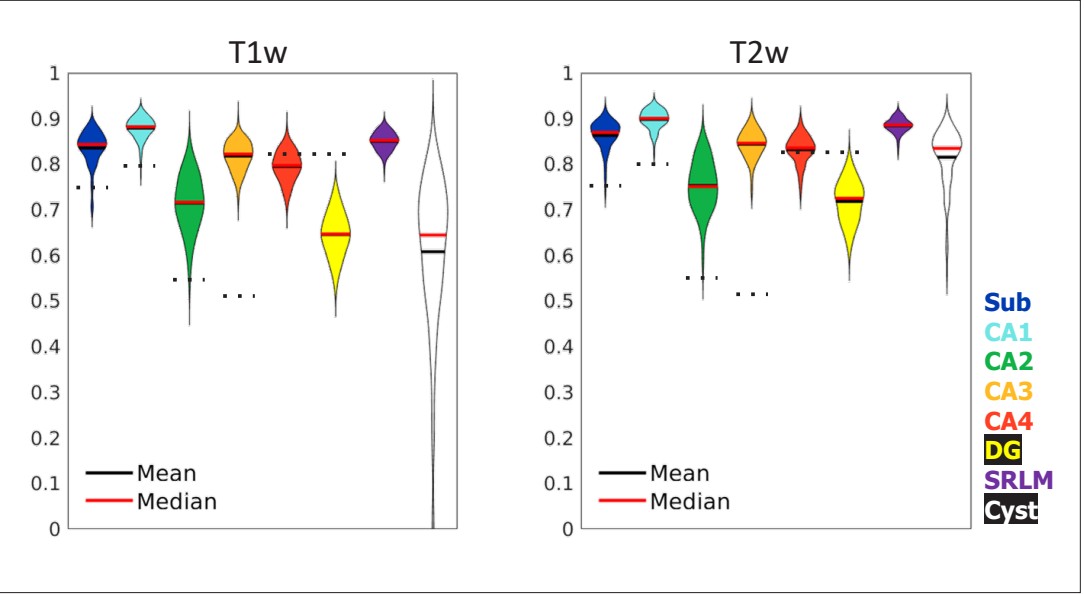

**Figure 5.** Test set performance in Dice overlaps between HippUnfold and manually unfolded subfields. All values are compared to ground truth manually defined tissues followed by unfolded subfield definition (manual unfold) to determine how small differences in grey matter parcellation propagate through the unfolding, subfield definition, and re-folding. Two models were trained in parallel using the same labels but different input MRI data modalities consisting of T1w or T2w data. Dotted black lines indicate corresponding values from *Yushkevich et al., 2015b*, who include stratum radiatum, lacunosum, and moleculaire (SRLM) in all labels and combine CA4 and DG into one label.

marginally poorer than T2w images which typically better show the SRLM and are popular in manual subfield segmentation protocols (*Yushkevich et al., 2015a*).

## Generalizability to unseen datasets and populations

We aimed to determine whether our pipeline would generalize to unseen datasets with different acquisition protocols and sample populations. Hippocampal morphometry, integrity, and subfields are often of interest in disease states where atrophy or other structural abnormalities are observed (*Thom, 2014*; *Haukvik et al., 2018*; *Steve et al., 2014*; *Carr et al., 2017*). For this reason, we examined the HCP-A datasets in which we anticipated cases of severe atrophy would be present in some older subjects. *Figure 5* shows results from one representative individual (an 80-year-old female with signs of age-related atrophy but good scan quality). Another common use-case for hippocampal subfield segmentation is on anisotropic T2w data which is considered optimal for performing manual segmentation in most protocols (*Yushkevich et al., 2015a*), but may impose challenges for our method due to the difference in resolution. We thus applied HippUnfold to 7T-TSE data and also illustrate one representative subfield segmentation result in *Figure 5*.

To demonstrate generalizability to pathological cases where hippocampal abnormalities can be confirmed, we also applied HippUnfold to a surgical epilepsy patient case. A 37-year-old female right-handed patient was investigated for surgical treatment of temporal lobe epilepsy, and clinical MR imaging at 1.5 T revealed a FLAIR hyper-intensity in the right hippocampus. The patient was imaged pre-surgically for a 7 T MRI research study, and the 0.7 mm MP2RAGE T1w (UNI-DEN) image was segmented using HippUnfold, which is shown in *Figure 5*. The patient underwent a right anterior temporal lobectomy and has been seizure-free (Engel class 1) for 4 years. Bilateral asymmetry is a strong indicator of epileptogenesis, and so results are examined for both the left and right hippocampi. Note that in addition to a loss in overall volume, the afflicted hippocampus showed relative sparing of CA2 which is a common observation in hippocampal sclerosis (*Blümcke et al., 2013*), as well as reduced digitations compared to the left hemisphere. Examining additional patients in future work may reveal whether morphometry could be a clinical marker of epileptogenesis in patients with no remarkable clinical lesions.

## Automated error flagging

Gold standard manual segmentations under the protocol used for subsequent unfolding were not available in novel datasets. Manually inspecting results from hundreds of subjects is time-consuming. We thus streamlined this process by flagging potential segmentation errors by examining Dice overlap with a more conventional segmentation approach: deformable registration. For all datasets described above, we applied deformable fast B-spline registration (*Modat et al., 2010*) to the corresponding T1w or T2w template. Tissue segmentation results (generated at the nnUNet stage) were then propagated to template space and overlap with standard template hippocampal masks were examined, which is shown in *Figure 5*. Any subject with a Dice overlap score of less than 0.7 was flagged and manually inspected for quality assurance. This made up 34/2126 (1.6%) samples in the HCP-YA T2w set (including training and testing subsets), 188/1312 (14.3%) samples from the HCP-A T2w set, 37/1312 (2.8%) samples from the HCP-A T1w set, and 3/92 (3.3%) samples from the 7T-TSE set. Closer

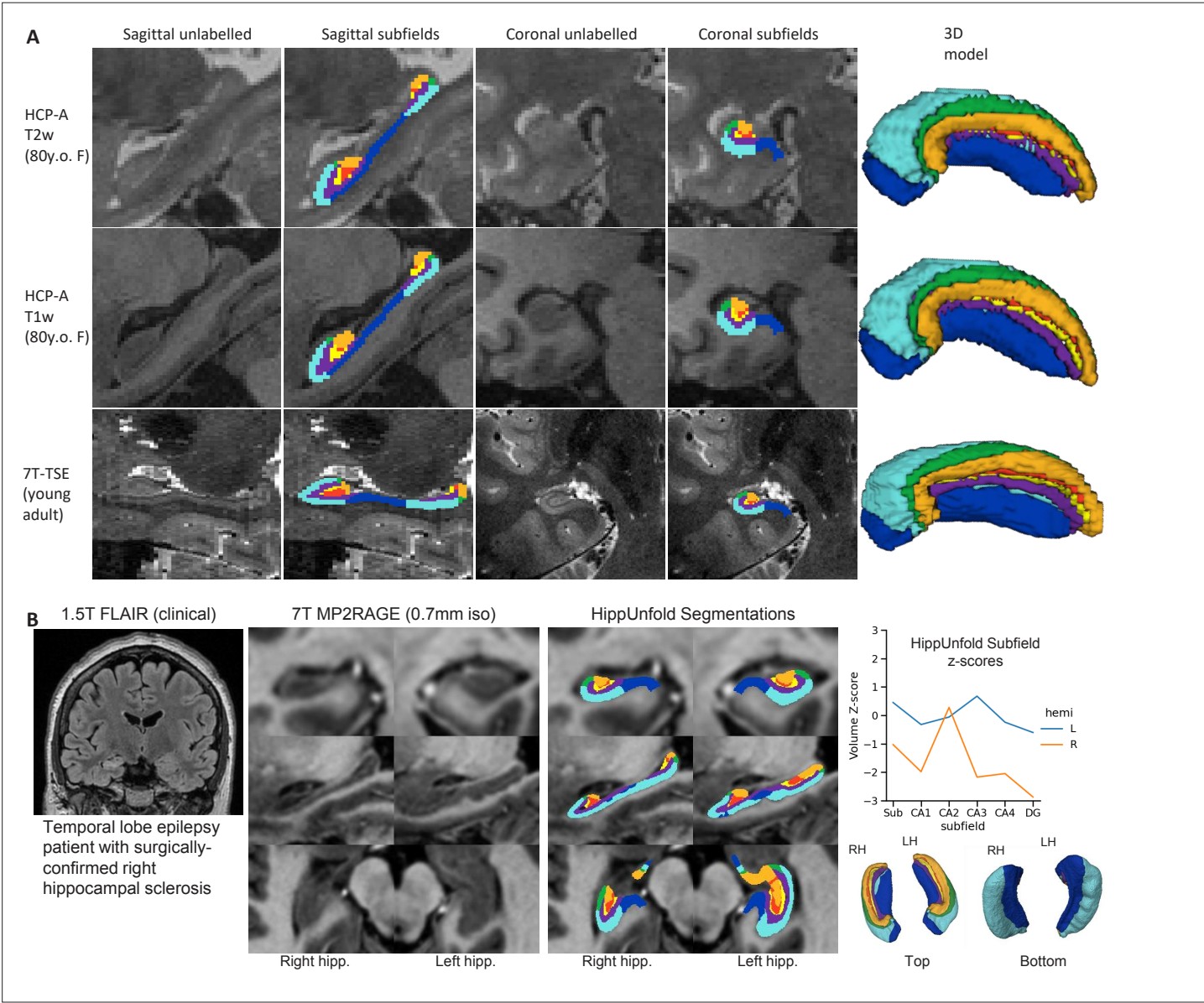

**Figure 6.** Examination of HippUnfold performance on additional datasets Human Connectome Project-Aging (HCP-A) (T1w and T2w) and anisotropic 7T-TSE data. (**A**) Sample subjects' HippUnfold subfield segmentation in native resolution. The first two rows come from the same subjects but using different input data modalities. (**B**) HippUnfold results from a 7 T MRI of a temporal lobe epilepsy patient with surgically confirmed right hippocampal sclerosis.

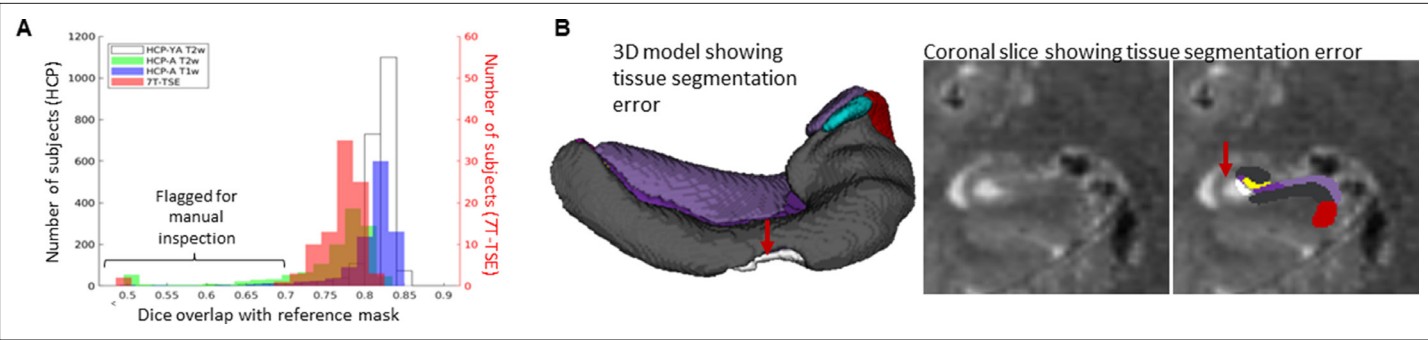

**Figure 7.** Automated error flagging via overlap with coarse, registration-based segmentation. (**A**) Subjects flagged for quality assurance from each dataset based on Dice overlap with a reference mask approximated via deformable registration. (**B**) Failed subject example illustrating missed tissue (red arrows) at the nnUNet stage of the HippUnfold pipeline.

inspection revealed that the vast majority of flagged cases were due to missed tissue in the nnUNet segmentation, an example of which is shown in *Figure 5*. It is interesting to note that the most flagged cases were seen in the HCP-A T2w dataset even though T2w is a popular acquisition protocol for hippocampal subfield segmentation (*Yushkevich et al., 2015a*; *Wisse et al., 2021*), and showed the best performance within the HCP-YA test set (*Figure 5*; *Figure 6A and B*). This was likely not due to the age of subjects since few of the HCP-A T1w were flagged as possible errors, but instead may have been due to T2w scan quality, which was observed to be poor in some subjects, causing poor definition of the outer hippocampal boundaries. We recommend that future users carefully inspect results from any flagged subjects, and cases with errors can be either discarded or manually corrected. Some work has already demonstrated that it is possible to synthesize or convert between MRI modalities (*Iglesias et al., 2021*), which could be used to alleviate the dependency on any single MR contrast. We cannot determine whether HippUnfold will work as intended on all new datasets, but within the generalization datasets examined here, results were excellent.

It is interesting to note that fewer failures were observed in HippUnfold using T1w data compared to T2w data (*Figure 7*), even though performance of nnUNet tissue classification were slightly higher with T2w images (*Figure 5*) and these are more common in hippocampal subfield imaging literature. Examining some failed cases, we see that these often had poor image quality, with subsequent issues like linear misregistration to the common CITI168 atlas or catastrophic nnUNet tissue classification failures.

## FAIR principles in development

We designed this pipeline to employ FAIR principles. As such, we have made use of several tools, conventions, and data standards to make HippUnfold extensible and easy to use.

The default file input-output structure of the HippUnfold command-line interface was built in compliance with the Brain Imaging Data Standards (BIDS) (*Gorgolewski et al., 2016*) Applications (BIDS-Apps) guidelines (*Gorgolewski et al., 2017*), and easily findable amongst the list of available BIDS Apps [https://bids-apps.neuroimaging.io/apps/]. This is achieved via Snakebids, a tool designed to interface between BIDS datasets and Snakemake (*Khan and Haast, 2021*). All aspects of HippUnfold use Snakemake (*Mölder et al., 2021*), a workflow management system based on Python which is reproducible, scalable, and seamlessly combines shell commands, Python code, and external dependencies in a human-readable workflow. There is no need to install these dependencies, which are containerized within the Singularity or Docker versions of HippUnfold.

Altogether, this means that in a single line this pipeline can be applied intelligently to any BIDS-complaint dataset containing a whole-brain T1w image or a T2w image (whole-brain or limited field of view) without having to specify further details. Typical runtimes on a standard desktop are 30–60 min per subject, but this is further parallelized for faster processing when multiple subjects and added compute resources (or cloud computing) are available. Additional flags can be used to extend functionality to many other use-cases, including T1w only, T2w only, diffusion-weighted imaging, cases where a manual tissue segmentation is already available, or ex vivo tissue samples.

Outputs of HippUnfold follow the standards for BIDS derivatives, and include preprocessed input images, volumetric subfield segmentations, inner, midthickness, and outer hippocampal surfaces, vertex-wise morphometric measures of thickness, curvature, and gyrification, and a brief quality control (QC) report. All surface-based outputs are combined into a Connectome Workbench (*Marcus et al., 2013*) specification file for straightforward visualization analogous to HCP neocortical reconstructions. Outputs can be specified to include images in the original T1w space or in the resampled, cropped space that processing is performed in.

All code, code history, documentation, and support are offered online (*Khan, 2022*) (https://github.com/khanlab/hippunfold).

## Discussion

One of the most powerful features of HippUnfold is its ability to provide topological alignment between subjects despite differences in folding (or digitation) structure. This is a critical element of mainstream neocortical analysis methods that, until now, has not been carried out systematically in the archicortex, or hippocampus. The power of this form of topological alignment is evident when mapping morphological or quantitative features across the hippocampus in a large population, which we demonstrate in *Figure 2*.

Segmentation of subfields is a task that is conceptually simplified through unfolding of the hippocampus to provide intrinsic anatomical axes. The axis we define as PD, which follows along the SLM in a coronal slice, is also a landmark relied upon in many manual segmentation protocols for the hippocampal subfields, including a histologically validated protocol that defines subfield boundaries by the proportional distance along the SLM (*Steve et al., 2017*). The head and tail are areas where these heuristics have conventionally been very difficult to apply, since the slice angulation optimal for the body is not optimal for the curved head and tail, and work using multiplanar reformatting provides one alternative for curved regions of the hippocampus (*Gross et al., 2020*). Our unfolding approach is conceptually analogous to these approaches, however, the added strength of our approach is that we apply the same conceptual rule (proportional distance along the SLM) while considering the entire 3D structure of the hippocampus.

We compare HippUnfold to other commonly used tools for hippocampal analysis, FS7 and ASHS (*Figure 4*). Both of these methods rely on smooth deformation of single or multi-atlas references, indicating they do not easily transfer to drastically different hippocampal folding patterns, which are often seen in the hippocampal head and tail across individuals. Both of these methods showed unfolded subfield patterns that were less consistent with ground truth histological literature than the output provided by HippUnfold. Common issues in other methods include introducing breaks in subfield topology, simplifications like the exclusion of the hippocampal tail, or inconsistent ordering of subfields. This highlights some of the advantages of HippUnfold, which was designed to overcome these issues explicitly.

Several factors make surface-based methods difficult to implement in the hippocampus, including its small size, and the difficulty of distinguishing the hippocampal sulcus or SRLM laminae that separate hippocampal folds. Here, we have overcome these issues using a highly generalizable and sensitive neural network 'U-Net' architecture, combined with our previously developed topological unfolding framework. Together, these methods achieved similar or better Dice overlap scores than what is typically seen between two manual raters on all subfields. We tested performance on new datasets ('generalization' datasets with different characteristics than the HCP training set) and saw good performance in nearly all cases. Specifically, we tested other common imaging protocols including different sample age groups (HCP-A) and thick-slice 7T-TSE acquisitions often used in targeted hippocampal subfield imaging (*Yushkevich et al., 2015a*). Though error rates were low, we do show how and why such errors sometimes occur, highlighting the importance that future users examine the brief QC reports included for each subject. Thus, while HippUnfold is shown to work well with all datasets examined here, we expect that the widespread adoption of higher-resolution acquisition techniques will further improve feasibility at other research institutes.

One important limitation of our method is that HippUnfold did not consistently show clear digitation in the hippocampal head, body, and tail which was sometimes seen in manual segmentation in the training set and in other work (see *Figure 4—figure supplement 1*). This reflects a lack of detail compared to histological ground truth materials, and affects downstream processing. That is, an

overly smoothed hippocampal surface will appear thicker and have a smaller surface area compared to one that captures the full extent of digitations. This smaller surface area also results in each subfield boundary being proportionally shifted. Future work could improve this pipeline by training and testing with higher-resolution data where digitations can more clearly be distinguished both in labelmaps and in the underlying images.

A single unfolded subfield atlas based on 3D BigBrain ground truth histology (*DeKraker et al., 2020*) was employed within HippUnfold for all subjects here. As illustrated in *Figure 4*, alternative unfolded subfield atlases can be used as well. Though previous work demonstrated reduced inter-individual variability of subfield boundaries in unfolded space (*DeKraker et al., 2018*), the extent to which subfield boundaries vary after unfolding is not yet known. In the neocortex, this issue is also present but partially mitigated with surface-based registration of available features like intracortical myelin, sulcal patterns, or thickness (e.g. *Glasser et al., 2016*). Such information could also be used in the unfolded hippocampus to further refine subject-specific subfield delineation, but would require histological ground truth data from multiple subjects to evaluate, ideally in 3D to avoid common out-of-plane sampling issues (*DeKraker et al., 2021*). This level of precision is likely beyond current typical MRI resolution levels, but should be investigated in future work aiming to combine in vivo and ex vivo or other more specialized imaging.

The current work has applications beyond subfield imaging, enabling new investigations of the hippocampus on a columnar and laminar scale. For example, rather than employing subfield ROI-based analyses, statistics can be performed on a per-vertex basis for vertices generated at different depths. This is in line with state-of-the-art neocortical analysis methods (*Van Essen et al., 2000*), and opens up the possibility of more precise localization of hippocampal properties. Similarly, it is worth noting that the methods used here are not necessarily restricted to MRI, as we have used the same surface-based unfolding in combination with manual segmentation to characterize the hippocampus in 3D BigBrain histology (*DeKraker et al., 2020*).

Altogether, we show that the BIDS App 'HippUnfold' that we have developed in this work (i) respects the different internal hippocampal folding configurations seen between individuals, (ii) can be applied flexibly to T1w or T2w data, sub-millimetric isotropic or thick-slice anisotropic data, and (iii) compares favourably to other popular methods including manual segmentation, ASHS and FS7. We believe this tool will open up many avenues for future work including examination of variability in hippocampal morphology which may show developmental trajectories or be linked to disease, or the examination of hippocampal properties perpendicular or tangential to its laminar organization with diffusion-weighted imaging. Finally, it is worth noting that the methods described here stand to improve existing techniques by providing greater anatomical detail and, critically, greater precision through topological alignment across individuals who vary in anatomical structure.

**Table 1.** MRI datasets used in training, evaluation, and comparison to extant methods. Methods employed include those proposed here (HippUnfold), the same processing but with manual segmentation (similar to previous work; *DeKraker et al., 2020*) (manual unfold), Freesurfer v7.2.0 (FS7) (*Iglesias et al., 2015*), and an atlas of manual segmentations (*Berron et al., 2017*) used in ASHS (*Yushkevich et al., 2015b*).

| Name | Modalities | Resolution | Sample size (L+R) | Methods employed |
|---|---|---|---|---|
| HCP-YA | T1w, T2w | 0.7 × 0.7 × 0.7 mm³ | n=590 (training) | HippUnfold<br>Manual unfold |
| | | | n=148 (testing) | HippUnfold<br>Manual unfold<br>FS7 |
| HCP-A | T1w<br>T2w SPACE<br>T2w TSE | 0.8 × 0.8 × 0.8 mm³<br>0.8 × 0.8 × 0.8 mm³<br>0.4 × 0.4 × 2.0 mm³ | n=1312 for T1w, T2w SPACE<br>n=200 for T2w TSE (FS7, ASHS)<br>n=200 for T1w (HippUnfold) | HippUnfold<br>FS7<br>ASHS |
| 7T-TSE (from ASHS atlas) | T2w | 0.4 × 0.4 × 1.0 mm³ | n=70 | HippUnfold<br>Manual subfields |

## Materials and methods

### Data

HippUnfold was designed and trained with the HCP 1200 young adult subject data release (HCP-YA) (*Van Essen et al., 2013*), and additionally tested on the HCP-A dataset (*Bookheimer et al., 2019*), and anisotropic (or thick-slice) 7T data (7T-TSE) from *Berron et al., 2017*, which is considered optimal by many hippocampal subfield researchers (*Yushkevich et al., 2015a*). Informed consent and consent to publish were obtained by the original authors of the open source data examined here. Each of the three datasets included research ethics board approvals, as well as informed consent and, in the HCP-A dataset, assessment of the subjects' ability to provide consent. For the single epilepsy patient case examined here, research ethics board approval and informed consent were collected at the Western University (HSREB # 108952, Lawson: R-17-156). These data are summarized briefly in *Table 1*.

### nnUNet training

U-Nets perform classification of each input image voxel, and it is not constrained by smooth displacements used in deformable atlas registration. This is important because smooth deformable registration can be ill-posed for an atlas with a different hippocampal folding configuration than the target. For example, when trying to register a hippocampus with two anterior digitations to one with four anterior digitations, topological breaks may be seen which leads to loss of detail and disproportionate stretching or compression of some subfields, an issue that is discussed in *DeKraker et al., 2021*. Instead, a U-Net classifies voxels individually based on a combination of local low-level and global high-level image features with no explicit smoothness constraints.

In the current work, gold standard training and test comparison segmentations were generated in a semi-automated but heavily supervised manner: a U-Net implementation (NiftyNet; *Gibson et al., 2018*, which is no longer maintained) was trained on existing data from *DeKraker et al., 2018*. This was then applied to new HCP-YA data and results were manually inspected. In many cases, results were poor due to the relatively small training sample size, but good quality segmentations from roughly 50% of subjects were selected and corrected by a manual rater (JD or MY) before being added to the initial training set for a new, de novo application of U-Net training. The process of inspection and manual correction was always performed according to the protocol outlined in *DeKraker et al., 2018*, to avoid systematic drift in rater performance. This process is typically referred to as incremental learning, and was applied in four iterations until a larger set of high quality, manually inspected and corrected segmentations (738 samples from 369 subjects) was achieved.

Once the gold-standard training data was obtained, we applied a U-Net implementation called nnUNet (*Isensee et al., 2021*). nnUNet was built to include many state-of-the art deep learning techniques including sensible hyperparameter selection, built-in fivefold cross-validation, and other features that have been shown to perform well and minimize possible sources of bias in medical imaging. We thus applied all default parameters in our use of this tool. Training was repeated using the same labelmaps but different underlying images for T1w, T2w, and DWI images. For each of these modalities, training took place on an NVIDIA T4 Turing GPU over 72 hr. Additional new models (or fine-tuned models) can also be trained and supplied within our code framework. Training data is available online at *Dekraker and Khan, 2022*.

### HippUnfold detailed pipeline

The command-line interface and options for HippUnfold are fully described online and in *Supplementary file 1*. A brief description of this pipeline is outlined here:

1. Preprocessing and resampling. Data is gathered via snakebids (*Khan and Haast, 2021*), which automatically and flexibly queries the specified BIDS directory for T1w and T2w images. Data is loaded and saved using NiBabel (*Brett et al., 2020*). Processing of each image is as follows:
   a. T1w: N4 bias correction is performed using the Advanced Normalization Toolkit (ANTs) (*Avants et al., 2008*) followed by affine registration (NiftyReg; *Modat et al., 2010*) to CITI168 atlas (*Pauli et al., 2018*). This transformation is composed (Convert 3D or c3d; *Yushkevich et al., 2019*) with a precomputed transform from CITI168 to oblique to the long axis of the hippocampus. Images are resampled to 0.3 mm$^3$ and cropped to 128 × 256 × 128 voxels

centred on the CITI168 left and right hippocampi. Left hippocampi are flipped sagittally to resemble right hippocampi. We refer to this as cropped coronal oblique space.

b. T2w: N4 bias correction is performed as above, and if multiple T2w images are present then they are rigidly registered (NiftyReg) and then averaged, a rudimentary form of super-resolution sampling (e.g. *Winterburn et al., 2013*). Rigid registration to the corresponding T1w image is then performed (NiftyReg), and resampled to cropped coronal oblique space as above.

A 'modality' flag is used to determine which image modalities should be used if multiple are present in the input BIDS directory. Within the HippUnfold code, optional flags can be used to skip preprocessing and registration. Manually segmented hippocampal tissues can also be specified, which can be useful in ex vivo MRI or other modalities on which the current nnUNet-based segmentation is not expected to work. In all cases, data are resampled to cropped coronal oblique space to match the nnUNet training setup. It is possible to skip this step only if a manually segmented hippocampal tissue class image is also provided (in which case nnUNet is not applied).

2. Tissue class segmentation. If a manually segmented hippocampal tissue image is not supplied, then the input image will be run through nnUNet (*Isensee et al., 2021*), a state-of-the-art implementation of a deep convolutional neural network (U-Net) designed for image segmentation (*Wiestler and Menze, 2020*; *Lu et al., 2017*). The output of nnUNet is a segmentation of tissue classes: hippocampal GM and the surrounding tissues which are used in defining unfolded coordinate boundaries: SRLM, medial temporal lobe cortex (MTLc), pial surface, hippocampal-amygdalar transition area (HATA), indusium griseum (IndGris), cysts, and the DG granule cell layer (which also makes up part of hippocampal GM but which marks an endpoint of the unfolding coordinate framework and so it was given a distinct label).

3. Post-processing. Here, we employed template shape injection (*Qiu and Miller, 2008*) to correct possible segmentation errors, making labelmaps more amenable to the previously developed hippocampal unfolding methods. The basic principle of template shape injection is to perform highly fluid deformable registration of a template segmentation labelmap to a given subject's segmentation labelmap. This differs from typical registration-based segmentation methods in that the registration is optimizing labels rather than image feature similarity (i.e. registration is performed with binarized and smoothed labels as multiple contrasts, rather than on MRI intensities). Specifically, we used mean squared error between labels as the cost function, which is minimized when identical labels are overlapping. In our implementation, we apply multi-contrast deformable registration using Greedy (*Yushkevich et al., 2019*). It should be noted that in principle this step is not necessary for our pipeline, but in practice it helps avoid possible errors due to nnUNet segmentation faults (see main text *Figure 5*).

The reference template that we applied was created using manual segmentations from an open source ex vivo dataset (*Wisse et al., 2017a*) that was manually segmented according to our previous manual hippocampal unfolding protocol (*DeKraker et al., 2018*). Labelmaps from 22 samples were combined using a standard template building ANTs script 'buildtem plateparallel.sh' (*Avants et al., 2008*). This template generation entails averaging all images and then registering each sample to the average, iteratively refining and sharpening the average image. This ex vivo dataset was selected for template building because we had high confidence in the quality of these segmentation since they contained higher resolution and contrast than other datasets while still including multiple samples.

4. Unfolding. This code is described in *DeKraker et al., 2018*, and was modified in *DeKraker et al., 2020*, but we will provide a short summary here.

## Intuition

Imagine we have a tangled piece of wire. We attach one end to something hot (100°C) and the other to something cold (0°C), and then wait for the temperature to equilibrate along the entire wire. We then have a second wire that is tangled up in a different pattern (and possibly with a different length). We attach the same hot and cold endpoints, and wait for it to equilibrate as well. Then, when we want to find topologically homologous points between the two wires, we find the spot where they have the same temperature, say 10°C (or 10% its length), or the same for any other pair of homologous points. This explanation works since the heat equation describing the equilibrium temperatures is the same as the Laplace equation if we assume that the heat conductance (or thermal diffusivity) is constant. These wires make up a 1D example, but the same principle also applies to a folded 2D sheet, where the endpoints are edges rather than ends. Here, we apply endpoints in two perpendicular directions: AP

(or HATA to ind.gris.) and PD (sub to DG), making up a standardized 2D indexing system (or 'unfolded space').

## Details

A Laplace field varying from 0 to 1 is generated across hippocampal GM, with 0 being at its anterior boundary with the HATA and 1 being at its posterior boundary with the IndGris (AP). This provides a scaled, smooth, geodesic way to index points along this axis. Another Laplace field is generated across the PD axis of the hippocampus (MTLc to DG), and together these two fields provide a coordinate system spanning hippocampus GM along two dimensions, which we plot as a flat rectangle (with a 2:1 aspect ratio to reflect the fact that the hippocampus is longer than it is wide). A third field is generated across the thickness of hippocampal GM (SRLM to outer boundary, or inner to outer, or IO). By default, the IO Laplace field is replaced by an equivolumetric model (*Waehnert et al., 2014*; *Huntenburg et al., 2018*), which helps account for the effects of curvature on laminar features (though this replacement can optionally be disabled). We then compute displacement fields for transforming each voxel from native space to the 'unfolded' space spanned by these three (AP, PD, and IO) fields, and vice versa.

Specifically, transformations for going between this unfolded space and native space are defined from Cartesian coordinates (x,y,z) to each Laplace field (AP, PD, and IO) for all hippocampal GM voxels. We performed piecewise linear interpolation (griddata from SciPy; *Virtanen et al., 2020*) to go from each unfolded coordinate (AP, PD, IO) to back to Cartesian coordinates (x,y,z). Rather than map Cartesian coordinates to Laplace coordinates ranging from 0 to 1 (as in previous work; *DeKraker et al., 2018*), we scale these gradients to make up a standard rectangular prism with a size of 256 × 128 × 16 voxels (dimensions corresponding to AP, PD, and IO, respectively), at a voxel size of 0.15625 mm$^3$ isotropic. This reference space is easily changed in the config file if a different unfolded resolution, size, or aspect ratio is desired. Each of these displacement fields is saved as a standard ITK 3D warp file in NIfTI format that can subsequently be applied to NIfTI or GIfTI files.

Unfolding of the DG is introduced in the current work. This is performed with the same methods described above but over the domain of the DG rather than all hippocampal GM. IO and PD fields are swapped with respect to the rest of the hippocampus reflecting the fact that during its development, the DG breaks from the rest of the cortical mantle and wraps around its terminus (CA4), making it topologically perpendicular to the rest of the hippocampus (*Duvernoy, 1998*). Endpoints for the DG are defined within the template shape used in step 3. Due to the thinness of the DG, it is often thinner than one voxel and so Laplace fields cannot easily be generated with the methods used in previous work. Thus, template shape injection is used to define the AP, PD, and IO fields within the DG, which were precomputed in the reference template with an idealized DG shape for unfolding. Thus, topological alignment between individuals does not perfectly follow the same Laplacian coordinate framework used in the rest of the hippocampus. Rather, this represents a more traditional volumetric approach to alignment via a template. The unfolded DG was defined by a rectangular prism with a size of 256 × 32 × 16, reflecting the fact that it is smaller than the rest of the hippocampus (PD) but still spans the same long (AP) axis.

5. Subfield definition. In previous work (*DeKraker et al., 2020*) we performed a highly detailed 3D ground truth segmentation of hippocampal subfields using 3D BigBrain histology (*DeKraker et al., 2020*). We mapped subfields using our Laplace coordinate framework, which provides implicit, topologically constrained registration between hippocampi. Thus, HippUnfold applies the same subfield boundary definitions to new samples in unfolded space, which are then propagated back to native space. Specifically, reference subfield labels already in unfolded space are warped to each subjects' native space using the warp files generated in step 4. Other unfolded subfield atlases can also be used, but BigBrain is the default since it is the most complete and detailed model of the hippocampal subfields to date.

6. GIfTI formatted outputs. In order to facilitate integration with other popular neuroimaging analysis tools, we have provided outputs in commonly used gifti surface formats in addition to volumetric nifti formats. Standardized unfolded surfaces corresponding to the inner, midthickness, and outer surface were generated for one standard unfolded template and propagated to each subjects' native, folded space using the warp files generated in step 4. Note that unfolded space is mapped to a rectangle rather than a sphere as is typically used in the neocortex, and so surfaces are not fully enclosed. Tessellation of vertices are available in a variety of densities categorized by their average vertex spacing in the native space: 0.5 mm (7262 vertices), 1 mm

(2004 vertices), 2 mm (419 vertices), or the legacy unfoldiso (32,004, ~32K, corresponding to the number of unfolded coordinates used in previous work, or 254×126).

Standardized unfolded tessellations were generated by starting with a 512×256 grid with each point connected to its neighbours, making a uniform mesh in unfolded space. Mesh vertices were iteratively removed until vertex distances after transforming to an averaged native space were achieved with the above spacings. In the case of the 32K surfaces, meshes were generated with 254×126 points with no vertices being removed, meaning that vertex distances are uniformly spaced in unfolded space but distorted in native space. In addition to hippocampal surfaces, DG surfaces are also generated, with the following unfolded meshes: 0.5 mm (1788 vertices), 1 mm (449 vertices), 2 mm (64 vertices), and unfoldiso (7620 vertices, 254×30).

7. Morphometry. Connectome Workbench commands (*Glasser et al., 2013*; *Marcus et al., 2011*) are used to extract measures of thickness between inner and outer surfaces, as well as curvature and gyrification along midthickness surfaces. The curvature metric is calculated using the mean curvature, calculated on a midthickness surface smoothed with the mean curvature midthickness surfaces, first smoothed by neighbourhood averaging (strength = 0.6, iterations = 100). The gyrification metric is defined as the ratio of native space surface area over unfolded space surface area, where the surface area is calculated at each vertex as the average of areas of connected triangles. Additional data (e.g. fMRI, DWI, or others) can be sampled at each vertex with the code provided in HippUnfold (the volume to surface mapping command in Connectome Workbench). With the implicit registration provided by unfolded space and the tessellation of these surfaces, such data can readily be compared across hippocampal samples without the need for further registration. These data can be subgrouped according to subfield labels, as in ROI analysis styles, or each vertex can be examined separately as in searchlight or data-driven analysis styles. Alternatively, gradient-based analyses can be applied based on Laplace coordinates and their corresponding surface mesh tessellations (see *Vos de Wael et al., 2018*, for example).

For even more implementation details, see *Supplementary file 4* – HippunFold algorithms.

## Acknowledgements

This work was supported by a Canadian Institutes for Health Research Project Grant (CIHR Grant # 366062) to AK and SK. AK was supported by the Canada Research Chairs program #950-231964, NSERC Discovery Grant #6639, and Canada Foundation for Innovation (CFI) John R Evans Leaders Fund project #37427, the Canada First Research Excellence Fund, and Brain Canada. JD was funded through a Natural Sciences and Engineering Research Council doctoral Canadian Graduate Scholarship (NSERC CGS-D). RAMH was supported by a BrainsCAN postdoctoral fellowship for this work.

Data were provided in part by the Human Connectome Project, WU-Minn Consortium (Principal Investigators: David Van Essen and Kamil Ugurbil; 1U54MH091657) funded by the 16 NIH Institutes and Centers that support the NIH Blueprint for Neuroscience Research; and by the McDonnell Center for Systems Neuroscience at Washington University.

Data and/or research tools used in the preparation of this manuscript were obtained from the National Institute of Mental Health (NIMH) Data Archive (NDA). NDA is a collaborative informatics system created by the National Institutes of Health to provide a national resource to support and accelerate research in mental health. Dataset identifier: 10.15154/1520707. This manuscript reflects the views of the authors and may not reflect the opinions or views of the NIH or of the Submitters submitting original data to NDA.

Data was also provided in part by *Berron et al., 2017*, in their published work 'A protocol for manual segmentation of medial temporal lobe subregions in 7 T MRI' (*Berron et al., 2017*) which includes MRI images and subfield segmentations.

# Additional information

## Funding

| Funder | Grant reference number | Author |
|---|---|---|
| Canadian Institutes of Health Research | 366062 | Stefan Köhler<br>Ali R Khan |
| Canada Research Chairs | 950-231964 | Ali R Khan |
| Natural Sciences and Engineering Research Council of Canada | RGPIN-2015-06639 | Ali R Khan |
| Canada Foundation for Innovation | 37427 | Ali R Khan |
| Brain Canada | Platform Support Grant for the Centre for Functional and Metabolic Mapping | Ali R Khan |
| Canada First Research Excellence Fund | BrainsCAN | Ali R Khan<br>Stefan Köhler |

The funders had no role in study design, data collection and interpretation, or the decision to submit the work for publication.

## Author contributions

Jordan DeKraker, Conceptualization, Resources, Data curation, Software, Formal analysis, Validation, Investigation, Visualization, Methodology, Writing – original draft, Writing – review and editing; Roy AM Haast, Validation, Visualization, Methodology, Writing – review and editing; Mohamed D Yousif, Resources, Data curation, Writing – review and editing; Bradley Karat, Conceptualization, Data curation, Methodology, Writing – review and editing; Jonathan C Lau, Conceptualization, Resources, Supervision, Funding acquisition, Validation, Project administration, Writing – review and editing; Stefan Köhler, Conceptualization, Resources, Data curation, Software, Supervision, Funding acquisition, Validation, Methodology, Project administration, Writing – review and editing; Ali R Khan, Conceptualization, Resources, Data curation, Software, Supervision, Funding acquisition, Validation, Methodology, Project administration, Writing – review and editing, Visualization, Writing – original draft

## Author ORCIDs

Jordan DeKraker ![ORCID] http://orcid.org/0000-0002-4093-0582
Roy AM Haast ![ORCID] http://orcid.org/0000-0001-8543-2467
Bradley Karat ![ORCID] http://orcid.org/0000-0002-6550-1418
Stefan Köhler ![ORCID] http://orcid.org/0000-0003-1905-6453
Ali R Khan ![ORCID] http://orcid.org/0000-0002-0760-8647

## Ethics

Informed consent and consent to publish were obtained by the original authors of the open source data examined here. Each of the three datasets included research ethics board approvals, as well as informed consent and, in the HCP-Aging dataset, assessment of the subjects' ability to provide consent. For the single epilepsy patient case examined here, research ethics board approval and informed consent were collected at the Western University (HSREB # 108952, Lawson: R-17-156).

## Decision letter and Author response

Decision letter https://doi.org/10.7554/eLife.77945.sa1
Author response https://doi.org/10.7554/eLife.77945.sa2

# Additional files

## Supplementary files

• Supplementary file 1. HippUnfold Documentation. This document fully describes the HippUnfold installation, command-line interface, options, outputs, and provides several useful pieces of

information including worked examples and useful tips on viewing data in other common platforms.

• Supplementary file 2. Side-by-side snapshot comparison of Human Connectome Project-Aging (HCP-A) segmentations results from HippUnfold, Freesurfer (FS7), and Automatic Segmentation of Hippocampal Subfields (ASHS) from the left hemisphere. Snapshots were taken at the conronal centroid, centroid + 15 slices, centroid + 30 slices, and the sagittal centroid.

• Supplementary file 3. Side-by-side snapshot comparison of Human Connectome Project-Aging (HCP-A) segmentations results from HippUnfold, Freesurfer (FS7), and Automatic Segmentation of Hippocampal Subfields (ASHS) from the right hemisphere. Snapshots were taken at the conronal centroid, centroid + 15 slices, centroid + 30 slices, and the sagittal centroid.

• Supplementary file 4. Detailed mathematical formulation of algorithms used throughout HippUnfold.

• Transparent reporting form

### Data availability

All code for the HippUnfold application has been made available at https://github.com/khanlab/hippunfold, (v1.2.0 release at https://zenodo.org/record/7063098). Data and code to generate the Figures shown in this study have been made available at https://zenodo.org/record/6360647. Training data for machine learning models have been made available at https://zenodo.org/record/7007362.

The following dataset was generated:

| Author(s) | Year | Dataset title | Dataset URL | Database and Identifier |
|---|---|---|---|---|
| Khan AR, DeKraker J | 2022 | HippUnfold HCP-YA Training Data | https://doi.org/10.5281/zenodo.7007362 | Zenodo, 10.5281/zenodo.7007362 |

The following previously published dataset was used:

| Author(s) | Year | Dataset title | Dataset URL | Database and Identifier |
|---|---|---|---|---|
| Bookheimer H | 2018 | Human Connectome Project - Aging | https://doi.org/10.15154/1520707 | Connectome Coordination Facility, 10.15154/1520707 |

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
