## [Editor Report]

This study presents a useful automated package called 'HippUnfold' in form of a BIDS App. The approach is solid and validated by comparing it against other methods in the field and has the potential to be used by a wide audience.

---

## [Decision Letter]

**Decision letter after peer review:**

Thank you for submitting your article "HippUnfold: Automated hippocampal unfolding, morphometry, and subfield segmentation" for consideration by *eLife*. Your article has been reviewed by 3 peer reviewers, and the evaluation has been overseen by a Reviewing Editor and Floris de Lange as the Senior Editor. The following individual involved in review of your submission has agreed to reveal their identity: Pierre-Louis Bazin (Reviewer #2).

Essential revisions:

Overall, this manuscript is well written, interesting, timely and will help resolve the debate in the field. We have the following suggestions to improve the manuscript:

1. As far as I understood, the U-Net approach defines individual landmarks needed for the unfolding and the unfolding provides a unique mapping between folded and unfolded space. However, if I understood correctly, in the next step the same subfield labels are enforced on every unfolded surface in the same way. My biggest concern is how we can be sure that these labels are valid for everyone.

2. Adding to this point, I find it very difficult to see any hippocampal structure in the 3T T1 data, for example HCP-YA case in figure 6 (but also HCP-A case in Figure 5 A, even true for the T2 image in the HCP-A case). This might be due to the image in the PDF and look better in the actual scan. However, as a human rater I would have no idea how to segment these cases and am wondering how the author's make sure that their approach produces a valid result and does not rely on the respective priors too much.

3. While the authors have addressed this in part by comparing the automated segmentation labels with the manual labels in young adults, there is no such data for populations that deviate more from young and healthy adults. Thus, there remains the question how their approach would deal with data from such populations.

4. I understand that manual segmentations are very tedious and labor intensive and might not be feasible in this project. However, maybe the authors could apply their pipeline to a dataset of a patient case with well-known abnormality and investigate the result? Alternatively, although the literature is less clear here, the authors could report on the differences that they see between HCP-A and HCP-YA on a group level and relate this to other findings in ageing or maybe even already existing work on these specific cohorts (in case these exist).

5. First, I would like to congratulate the authors in building an elegant toolbox for hippocampal analysis, with many valuable features for the basic and advanced users alike. I expect the points I raised can be addressed by reframing the presentation of the software, focusing more on what it provides in terms of a representation and less on whether or not it provides a better subfields definition.

6. My first point above should be addressable by including all algorithmic details either in the main text or in an appendix, so the article is self-contained with regard to the methodology. Details of the UNet preprocessing and architecture, Laplacian coordinate mapping algorithm, and morphometric feature extraction should be included. The 'Hippunfold detailed pipeline' section remains vague: concrete descriptions including mathematical formulas and algorithm parameters would be better. A figure showing the outputs of the different steps would also be helpful.

7. The second point requires carefully re-evaluating the claims made about topology, and separating the unfolding and labeling question. In the end, the provided algorithm does not perform any subfield labeling of individual hippocampi, but only transfers a fixed label map from BigBrain onto individual anatomies, using the unfolding coordinates as a proxy. Thus the results of Figure 4 are misleading, since they compare the quality of the unfolding and not the labeling. This point should be made clear, and the comparisons of Figure 3 need to be altered, maybe discussing rather the limited variability of the unfolded labels from ASHS and FreeSurfer as an indication of the quality of the unfolded representation. If the authors want to compare the quality of subfield labelings across methods, those comparisons should be done in voxel space. Note also that the claim that ASHS and FreeSurfer do not preserve topology is unnecessary and debatable (e.g. internally the FreeSurfer algorithm uses a fixed-topology mesh, so it does preserve its own definition of topology).

8. Here I would rather have a comparison with other representations, e.g. using the volumetric space directly, mapping to the outer surface, or defining a medial axis representation. Why is mapping hippocampal information onto a 2D plane better? Does this preserve more the common features across hippocampi than other options? While this idea is hinted at when discussing variability in folding, it could be empirically tested.

9. This also links to the third question of implicit alignment which could be tested for instance by inspecting the variation in subfield boundaries from volumetric methods in the experiment of Figure 3. Note also that features mapped onto the unfolded representation of Figure 2 could be co-registered into a 2D atlas, and the corresponding deformations could be evaluated.

10. Another question related to representation is the decision to use a rectangular map rather than a more irregular one similar to those used in cortical and cerebellar flat maps: by doing so, some of the regions get a distorted importance (as shown in the mesh maps presented in the documentation). It would be good to provide a measure of the distortion to be expected.

11. Finally, while the software and documentation are very well organized, I was unable to run the app on the test data folders or my own data, using docker, singularity or the poetry installation option, which is absolutely required to complete this review. The 'getting started' section should also include a full processing script example on a test data set, outlining the main steps and basic parameters, especially as the toolbox is quite flexible and thus quite complex. Commands used for visualizing the various results should also be given in the 'Outputs' section, so users can visualize their data as in the examples. Given the richness of the manual delineation and segmentation effort, it would be valuable to release the training and testing data openly (note also that it is quite important for the U-Net step, where the training set properties have a strong impact on performance and potential biases).

12. In general, the paper is well written, but there are multiple areas that I have some issues with following the logical flow of what is being proposed. For example, the paper begins with demonstrating multiple metrics that are projected onto the hippocampal flatmap that includes thickness, myelin, curvature, gyrification, etc. It is unclear as to what information the authors want to convey here. This is the first mention of many of these multiple metrics as well and therefore their relevance is ultimately not extremely clear. As a result, it is hard to support their claim that "differences in morphological and quantitative features can be seen across the hippocampus, particularly across the subfields" as the goals of this particular figure are not at all clear.

13. Line 147: It is not totally accurate to state that ASHS makes use of multi-atlas registration as it also uses AdaBoost to correct for segmentation inaccuracies.

14. For the FreeSufer and ASHS comparisons – is it possible to provide some quantification of errors or anything like that? I think it would be helpful to quantify the differences in a more accurate manner. If this is in a previous publication and I missed it, it could be useful to reiterate here. The qualitative difference is nice – but there is room to compare them more quantitatively to one another.

15. For the validation of the U-NET, details on the manual segmentation protocol, who did it, and its reliability are crucial. Training/testing paradigms would be helpful here. So would Bland-Altmann plots. I think in general the validation of these segmentations is quite poor – so more metrics that demonstrate the segmentation beyond dice overlaps would be helpful.

16. It is unclear how generalizable the method is outside of HCP acquisitions.

*Reviewer #1 (Recommendations for the authors):*

As far as I understood, the U-Net approach defines individual landmarks needed for the unfolding and the unfolding provides a unique mapping between folded and unfolded space. However, if I understood correctly, in the next step the same subfield labels are enforced on every unfolded surface in the same way. My biggest concern is how we can be sure that these labels are valid for everyone.

Adding to this point, I find it very difficult to see any hippocampal structure in the 3T T1 data, for example HCP-YA case in figure 6 (but also HCP-A case in Figure 5 A, even true for the T2 image in the HCP-A case). This might be due to the image in the PDF and look better in the actual scan. However, as a human rater I would have no idea how to segment these cases and am wondering how the author's make sure that their approach produces a valid result and does not rely on the respective priors too much.

While the authors have addressed this in part by comparing the automated segmentation labels with the manual labels in young adults, there is no such data for populations that deviate more from young and healthy adults. Thus, there remains the question how their approach would deal with data from such populations.

I understand that manual segmentations are very tedious and labor intensive and might not be feasible in this project. However, maybe the authors could apply their pipeline to a dataset of a patient case with well-known abnormality and investigate the result? Alternatively, although the literature is less clear here, the authors could report on the differences that they see between HCP-A and HCP-YA on a group level and relate this to other findings in ageing or maybe even already existing work on these specific cohorts (in case these exist).

*Reviewer #2 (Recommendations for the authors):*

First, I would like to congratulate the authors in building an elegant toolbox for hippocampal analysis, with many valuable features for the basic and advanced users alike. I expect the points I raised can be addressed by reframing the presentation of the software, focusing more on what it provides in terms of a representation and less on whether or not it provides a better subfields definition.

My first point above should be addressable by including all algorithmic details either in the main text or in appendix, so the article is self-contained with regard to the methodology. Details of the UNet preprocessing and architecture, Laplacian coordinate mapping algorithm, and morphometric feature extraction should be included. The 'Hippunfold detailled pipeline' section remains vague: concrete descriptions including mathematical formulas and algorithm parameters would be better. A figure showing the outputs of the different steps would also be helpful.

The second point requires carefully re-evaluating the claims made about topology, and separating the unfolding and labeling question. In the end, the provided algorithm does not perform any subfield labeling of individual hippocampi, but only transfers a fixed label map from BigBrain onto individual anatomies, using the unfolding coordinates as a proxy. Thus the results of Figure 4 are misleading, since they compare the quality of the unfolding and not the labeling. This point should be made clear, and the comparisons of Figure 3 need to be altered, maybe discussing rather the limited variability of the unfolded labels from ASHS and FreeSurfer as an indication of the quality of the unfolded representation. If the authors want to compare the quality of subfield labelings across methods, those comparisons should be done in voxel space. Note also that the claim that ASHS and FreeSurfer do not preserve topology is unnecessary and debatable (e.g. internally the FreeSurfer algorithm uses a fixed-topology mesh, so it does preserve its own definition of topology).

Here I would rather have a comparison with other representations, e.g. using the volumetric space directly, mapping to the outer surface, or defining a medial axis representation. Why is mapping hippocampal information onto a 2D plane better? Does this preserve more the common features across hippocampi than other options? While this idea is hinted at when discussing variability in folding, it could be empirically tested.

This also links to the third question of implicit alignment which could be tested for instance by inspecting the variation in subfield boundaries from volumetric methods in the experiment of Figure 3. Note also that features mapped onto the unfolded representation of Figure 2 could be co-registered into a 2D atlas, and the corresponding deformations could be evaluated.

Another question related to representation is the decision to use a rectangular map rather than a more irregular one similar to those used in cortical and cerebellar flat maps: by doing so, some of the regions get a distorted importance (as shown in the mesh maps presented in the documentation). It would be good to provide a measure of the distortion to be expected.

Finally, while the software and documentation are very well organized, I was unable to run the app on the test data folders or my own data, using docker, singularity or the poetry installation option, which is absolutely required to complete this review. The 'getting started' section should also include a full processing script example on a test data set, outlining the main steps and basic parameters, especially as the toolbox is quite flexible and thus quite complex. Commands used for visualizing the various results should also be given in the 'Outputs' section, so users can visualize their data as in the examples. Given the richness of the manual delineation and segmentation effort, it would be valuable to release the training and testing data openly (note also that it is quite important for the U-Net step, where the training set properties have a strong impact on performance and potential biases).

*Reviewer #3 (Recommendations for the authors):*

– Clear definition of goals and the novelty of the work.

– Better comparison against other methods.

– Better comparison against manual segmentation (needs more than just the Dice).

– Lack of demonstration of generalizability. Need to see how this may work in the context of other data acquisition streams.

[Editors' note: further revisions were suggested prior to acceptance, as described below.]

Thank you for resubmitting your work entitled "Automated hippocampal unfolding for morphometry and subfield segmentation using HippUnfold" for further consideration by eLife. Your revised article has been evaluated by Floris de Lange (Senior Editor) and a Reviewing Editor.

The manuscript has been improved but there are some remaining issues that need to be addressed, as outlined below:

1. The description of some of the central methods used in the article (Laplacian embedding, shape injection) is too limited to understand fully how we obtain the unfolding. I see the point of 'increased complexity with increasing depth', but the article does not reach the level where the algorithms are explicitly described. It is also unclear if the authors used third-party software or their own implementation of these two methods.

2. A second remaining issue I have is the somewhat puzzling fact that the T1w trained version of hippunfold performed better than the T2w one for the HCP-aging dataset: it would be good to understand why that is the case.

3. Finally, I still could not run the algorithm successfully on the provided example. Both Docker and Singularity provide very little information about why they fail. Installing and running the development version almost works, but only after installing several additional packages and non-standard research software from other groups (connectome workbench, NiftyReg, c3d...). None of these dependencies are described in the documentation. I would recommend the authors test their installation procedure on a bare-bones OS, for instance in a Linux virtual machine, as it is often challenging to remember what elements of a customized installation are being used or not. I am confident that the remaining issues are small, and that the usability of the software will be increased in the exercise.

*Reviewer #2 (Recommendations for the authors):*

I thank the authors for a thorough revision with additional validation experiments, which generally addresses my concerns and issues. I particularly appreciate the addition of the FreeSurfer and ASHS/Magdeburg labelings as options: it clarifies the separation between unfolding and labeling, and also provides continuity for users who have worked with these labels in previous studies. I also commend the authors for releasing their training data, increasing transparency, and for providing actual test data sets.

However, I would still argue that the description of some of the central methods used in the article (Laplacian embedding, shape injection) is too limited to understand fully how we obtain the unfolding. I see the point of 'increased complexity with increasing depth', but the article does not reach the level where the algorithms are explicitly described. It is also unclear if the authors used third-party software or their own implementation of these two methods.

A second remaining issue I have is the somewhat puzzling fact that the T1w trained version of hippunfold performed better than the T2w one for the HCP-aging dataset: it would be good to understand why that is the case.

Finally, while these two issues above are minor, I still could not run the algorithm successfully on the provided example. Both Docker and Singularity provide very little information about why they fail. Installing and running the development version almost works, but only after installing several additional packages and non-standard research software from other groups (connectome workbench, NiftyReg, c3d...). None of these dependencies are described in the documentation. I would recommend the authors to test their installation procedure on a bare-bones OS, for instance in a Linux virtual machine, as it is often challenging to remember what elements of a customized installation are being used or not. I am confident that the remaining issues are small, and that the usability of the software will be increased in the exercise.

---

## [Author Response]

Essential revisions:Overall, this manuscript is well written, interesting, timely and will help resolve the debate in the field. We have the following suggestions to improve the manuscript:1. As far as I understood, the U-Net approach defines individual landmarks needed for the unfolding and the unfolding provides a unique mapping between folded and unfolded space. However, if I understood correctly, in the next step the same subfield labels are enforced on every unfolded surface in the same way. My biggest concern is how we can be sure that these labels are valid for everyone.

Yes, the unfolding provides a unique mapping for each hippocampus between native (folded) and unfolded space, which allows an atlas defined in the unfolded space to be mapped to any hippocampus. There are a number of reasons, both conceptually and empirically why we believe this approach is a valid one:

a. The landmarks we use decompose the hippocampus along intrinsic anatomical axes of the hippocampus. The axis we define as proximal-distal (PD), which follows along the SLM in coronal slice, is also a landmark relied upon in many manual segmentation protocols for the hippocampal subfields. As one example, Steve et al. (2017) developed a histologically-validated protocol that defined subfield boundaries by the proportional distance along the SLM in a coronal slice through the body of the hippocampus. Our unfolding approach is conceptually analogous to this approach, since a vertical line through the unfolded space at a mid-body slice will similarly describe subfield boundaries as a proportional distance along the SLM. Protocols developed for segmenting the body of the hippocampus on MRI that are based on geometric heuristics can also be seen as analogous to our approach. For example, the proposed Hippocampal Subfield Group harmonized protocol (protocol still under community review, as yet unpublished), that defines boundaries based on angles relative to axes drawn on the hippocampus, is also conceptually using the proportional distance (or arc length) along the SLM if we assume a circular shape is used to model the SLM.

Steve TA, Yasuda CL, Coras R, Lail M, Blumcke I, Livy DJ, Malyhkin N, Gross DW. (2017) Development of a histologically validated segmentation protocol for the hippocampal body. NeuroImage, 157, 219-232.

b. The added strength of our approach over these analogous methods (applied in the body) is that we apply the same conceptual rule (proportional distance along the SLM) as a means to segment the head and tail as well. The head and tail are areas where the heuristics have conventionally been very difficult to apply, since the slice angulation optimal for the body is not optimal for the curved head and tail; in certain regions, a slice perpendicular to the body is the optimal one. This phenomenon has also been observed by others (Gross et al., 2020), and has even been used to more accurately apply the known heuristics to the more curved regions of the hippocampus anterior and posterior. We have added a new paragraph to the discussion to provide a succinct overview of these points.

Gross D, Misaghi E, Steve TA, Wilman AH, Beaulieu C. (2020) Curved multiplanar reformatting provides improved visualization of hippocampal anatomy. Hippocampus 30(2), 156-161.

c. Empirical validity of the unfolding approach could also be assessed by demonstrating that the same subfields defined independently on multiple subjects/hippocampi map to the same regions in the unfolded space. We have performed this validation in our initial paper that described our unfolding approach (Dekraker et al., 2018). Here, manually segmented subfields from multiple subjects were mapped consistently to the unfolded space, as demonstrated by 1) depiction of manual subfields from all individuals mapped to the unfolded space with the standard error in subfield boundaries visualized (Figure 5C), and 2) Dice scores using a leave-one-out approach to segment each subject with an average atlas of the other subjects (red/blue bars in Figure 5). Note that this method made use of our semi-automated tool for unfolding the hippocampus, and the codebase for our toolset has evolved considerably since. Thus, we have also included an updated analogous experiment with our current HippUnfold tool on a larger set of hippocampi, and manual segmentations from a completely independent rater and dataset. This is shown in Figure 4 – Supplement 1, specifically the panel in the top right that depicts the manual subfield boundaries of N=70 hippocampi (35 subjects) after being transformed to the unfolded space. Similar figures for subfields from existing automated methods (ASHS, Freesurfer) are also shown in Figure 4 of the manuscript. These all demonstrate that the boundaries between subfields have a high level of agreement, with much of the inter-subject variability accounted for in the unfolding.

DeKraker, Jordan, Kayla M. Ferko, Jonathan C. Lau, Stefan Köhler, and Ali R. Khan. "Unfolding the hippocampus: An intrinsic coordinate system for subfield segmentations and quantitative mapping." *Neuroimage* 167 (2018): 408-418.

d. In our approach, the same atlas is mapped to every subject; we have used a manual segmentation from the BigBrain hippocampi since it was (and currently still is) the highest resolution 3D reference that exists for the hippocampus. However, it is still a single subject, and thus no anatomical variability is encoded in this labelling. We have now provided alternatives to this single-subject subfield atlas, generated by mapping many segmented individual hippocampi to the unfolded space to derive a probabilistic labelling of each subfield. Maximum-probability label maps of the Magdeburg atlas (35 subjects released as a 7T ASHS atlas) along with Freesurfer segmentations of the same subjects were generated using companion scripts in http://github.com/khanlab/hippunfold-create-atlas (main and freesurfer branches respectively). These are now available in the HippUnfold tool as command-line options. Note that although these atlases will provide labels that are generally consistent with ASHS and Freesurfer, they still could differ substantially since the gray-matter and SRLM segmentations are defined by our trained U-net. These atlases are as seen in Figure 4 of the manuscript.

e. Finally, as evident by the probabilistic nature of subfields mapped to the unfolded spaces, we are aware that there is still some residual variability that our unfolding does not account for. One could potentially to account for this remaining variability by performing a subsequent non-linear registration in the unfolded space. However, implementation of this, including the optimal features to drive this registration and how much to regularize are choices that need to be further explored in future work. We have added a new paragraph in the Discussion section detailing this possibility.

2. Adding to this point, I find it very difficult to see any hippocampal structure in the 3T T1 data, for example HCP-YA case in figure 6 (but also HCP-A case in Figure 5 A, even true for the T2 image in the HCP-A case). This might be due to the image in the PDF and look better in the actual scan. However, as a human rater I would have no idea how to segment these cases and am wondering how the author's make sure that their approach produces a valid result and does not rely on the respective priors too much.

We have added Figure 3 to more clearly show the visible internal hippocampal structures like the SRLM and digitations. Note that the out-of-plane issue discussed above is a major reason these scans are difficult to interpret, and is not an issue for U-Net automated segmentation.

Note also that one advantage that U-Net has over a human rater is that it makes use of fully 3D convolutional layers, meaning that it is sensitive to multiple slices and slice directions in concert. The advantage of this can be seen when interactively scrolling through volumes, but is hard to capture in a static image. For example, digitations in the hippocampal head are often only clear in the coronal plane, while digitations in the body and tail of the hippocampus are often only evident in the sagittal plane. The same is true of the SRLM, which enters into the ‘crease’ of each digitation, making it hard to see in some ‘out-of-plane’ slices at any resolution (but especially at lower resolutions). We discuss some of these issues in our recent Opinion paper (DeKraker et al., 2021), but wanted to avoid these more technical discussions in the present manuscript.

It is also true that U-Net relies on a prior distribution in the absence of clear image features, and indeed, this can produce overly smooth hippocampal segmentations when definitive digitations are not visible. This is certainly a limitation and is discussed at length (Discussion section). We feel the current HippUnfold behaviour of relying on priors in the absence of detailed image features, and flagging subjects that show possible catastrophic failures, is desirable in most large-scale neuroimaging applications.

3. While the authors have addressed this in part by comparing the automated segmentation labels with the manual labels in young adults, there is no such data for populations that deviate more from young and healthy adults. Thus, there remains the question how their approach would deal with data from such populations.

We have now included additional analyses and visual examples from a convenience sample of 100 subjects from the HCP-aging dataset, with ages that range from 37 to 87 years of age, comparing subfield segmentations and volumes using HippUnfold against Freesurfer and ASHS (Figure 3). The HCP-aging dataset MRI protocol includes a high-resolution hippocampal T2 TSE sequence, and since this sequence depicts coronal slices of the hippocampus in greatest detail, we used these images as inputs to Freesurfer and ASHS. For HippUnfold we used the T1w image and corresponding T1w trained model since it provides the most robust outputs, but we have registered the outputs to the T2 TSE space for visual comparison against the other approaches. The comparisons of subfield volume demonstrate a generally high level of agreement across methods, and similar age-related declines in volume. Note that since the subfield boundary definitions differ across segmentation methods, absolute agreement was not expected. The visual comparisons between volumetric segmentations (e.g. a single subject shown in Figure 3), demonstrates anatomical accuracy of the proposed approach. Comparisons for all 100 subjects are shown in Figure 3 and Figure 3 – Supplement 1, and qualitatively demonstrates the robustness of the T1w HippUnfold model.

4. I understand that manual segmentations are very tedious and labor intensive and might not be feasible in this project. However, maybe the authors could apply their pipeline to a dataset of a patient case with well-known abnormality and investigate the result?

We have now added results on an epilepsy patient case as an example where the hippocampal pathology is well-characterized. A 37 year-old female right-handed patient was investigated for surgical treatment of temporal lobe epilepsy, and clinical MR imaging at 1.5T revealed a FLAIR hyper-intensity and volume loss in the right hippocampus consistent with a radiological diagnosis of mesial temporal sclerosis. The patient was imaged pre-surgically for a 7 Tesla MRI research study, and the 0.7mm MP2RAGE T1w (UNI-DEN) image was segmented using HippUnfold. Figure 6B demonstrates a unilateral volumetric reduction in all subfields except for CA2, characteristic of classical mesial temporal sclerosis. The patient underwent a right anterior temporal lobectomy and has been seizure-free (Engel class 1) for 4 years.

Alternatively, although the literature is less clear here, the authors could report on the differences that they see between HCP-A and HCP-YA on a group level and relate this to other findings in ageing or maybe even already existing work on these specific cohorts (in case these exist).

We have been investigating lifespan changes in hippocampal structure and function in separate ongoing work (see the OHBM abstract listed below), using HippUnfold to examine morphological differences from the Aging and Development HCP datasets. Indeed, early results showed thinning and reduced surface areas in older adults, which was presented at OHBM 2022. As detailed in response 3 above, we have also now added a comparison of 100 HCP-A subjects with extant Freesurfer and ASHS methods as well.

**“**Unfolding the human hippocampal lifespan: changes in morphology and functional connectivity” at: Annual Meeting of the Organization of Human Brain Mapping, Glasgow, UK, 2022 (international)

5. First, I would like to congratulate the authors in building an elegant toolbox for hippocampal analysis, with many valuable features for the basic and advanced users alike. I expect the points I raised can be addressed by reframing the presentation of the software, focusing more on what it provides in terms of a representation and less on whether or not it provides a better subfields definition.6. My first point above should be addressable by including all algorithmic details either in the main text or in an appendix, so the article is self-contained with regard to the methodology. Details of the UNet preprocessing and architecture, Laplacian coordinate mapping algorithm, and morphometric feature extraction should be included. The 'Hippunfold detailed pipeline' section remains vague: concrete descriptions including mathematical formulas and algorithm parameters would be better. A figure showing the outputs of the different steps would also be helpful.

Thank you, we have now included additional information, especially in the “detailed pipeline” section. We would like to note that the project generally aimed to the follow the principle of increased complexity with increased depth: the main body of the text should be intuitively comprehensible for a broad audience, the detailed pipeline for people familiar with image segmentation, registration, and surface mapping methods, and the online documentation should fully describe all possible parameter choices. Note that we reuse many existing tools for segmentation (eg. UNet) and registration (eg. ANTs, wb_command) which are described in other, referenced works. The Laplace coordinate system is a very unique component, and we have published another paper where we fully describe and utilize this method with manual segmentation (DeKraker et al., 2018; updates in 2019). We have now included some additional descriptions from these papers here, as well as generally adding more descriptions throughout.

7. The second point requires carefully re-evaluating the claims made about topology, and separating the unfolding and labeling question. In the end, the provided algorithm does not perform any subfield labeling of individual hippocampi, but only transfers a fixed label map from BigBrain onto individual anatomies, using the unfolding coordinates as a proxy.

Yes, and we refer to Response 1 for evidence of why we believe this approach is a valid one, both conceptually and empirically.

Thus the results of Figure 4 are misleading, since they compare the quality of the unfolding and not the labeling. This point should be made clear, and the comparisons of Figure 3 need to be altered, maybe discussing rather the limited variability of the unfolded labels from ASHS and FreeSurfer as an indication of the quality of the unfolded representation. If the authors want to compare the quality of subfield labelings across methods, those comparisons should be done in voxel space.

You are right that Figure 3 and 4 depend on the quality of unfolding, and this quality is perhaps the most critical component of the work shown here. The only available reference or gold standard to which we can compare unfolding quality is using manually defined hippocampal gray matter and surrounding labels, which was our comparison for the Dice scores in Figure 5. Note that these comparisons were all made in native space on a voxel wise basis. If we include only gray matter without parcellating it into subfields, Dice scores were >0.95 since this tissue is large and Dice scores depend on the sizes of the regions being compared. We thus calculated Dice scores on the subfield level in order to determine how small differences in gray matter parcellation propagate through the unfolding, subfield definition, and re-folding. Calculating Dice scores of the individual subfields also makes this evaluation more comparable to other methods used in the literature.

We have also now added a comparison of HippUnfold, ASHS, and FS subfield pipelines in Figure 3 and Figure 3 – Supplement 1, but this analysis reveals only that these methods differ, and not which method shows best correspondence to ground truth histology observations.

We have now also included some discussion of variability of subfield boundaries in unfolded space. We acknowledge this is a limitation, though it is not unique to surface-based parcellation in the hippocampus but rather is also an issue in surface-based neocortical parcellation.

Note also that the claim that ASHS and FreeSurfer do not preserve topology is unnecessary and debatable (e.g. internally the FreeSurfer algorithm uses a fixed-topology mesh, so it does preserve its own definition of topology).

You are right that Freesurfer preserves its own intrinsic topology, however, it does not have the topological ordering of subfields which has been shown consistently in histology (eg. CA1 should never directly border CA3; all subfields should be present through the full longitudinal extent of the hippocampus). Additionally, histology and other work reveals that the underlying tissue of the hippocampus is folded but Freesurfer labels often stretch or compress multiple folds/digitations together which, when ideally unfolded, leads to discontinuities on both the subject and the average level (Figure 4). The same is true of ASHS, though to a lesser extent and not in the average as unfolded via HippUnfold. We have clarified this sense in which we use the term ‘topology’ in text to avoid confusion.

8. Here I would rather have a comparison with other representations, e.g. using the volumetric space directly, mapping to the outer surface, or defining a medial axis representation. Why is mapping hippocampal information onto a 2D plane better? Does this preserve more the common features across hippocampi than other options? While this idea is hinted at when discussing variability in folding, it could be empirically tested.

Although this is an interesting suggestion, we believe that a comprehensive comparison of representations would be outside of the scope of this work. The conceptual advantages are mainly gained through the use of hippocampal-centric coordinates, defined by hippocampal anatomical landmarks, and not necessarily in their projection to a 2D plane. A review of the conceptual advantages to the unfolding approach is described in Response 1, and these relate mainly to inter-subject alignment and subfield segmentation. The sensitivity and specificity of morphometric analyses using different hippocampal representations would require further study to evaluate what differences may exist.

9. This also links to the third question of implicit alignment which could be tested for instance by inspecting the variation in subfield boundaries from volumetric methods in the experiment of Figure 3. Note also that features mapped onto the unfolded representation of Figure 2 could be co-registered into a 2D atlas, and the corresponding deformations could be evaluated.

Yes, a deformation-based morphometry approach applied in the 2D atlas space would provide quantitative estimates of how much subfield variation is accounting for via unfolding/alignment. This analysis, however, is less anatomically-meaningful when the subfield boundaries are defined based on manual segmentation heuristics, rather than the underlying ground-truth anatomy. We have added a Discussion section on this issue.

10. Another question related to representation is the decision to use a rectangular map rather than a more irregular one similar to those used in cortical and cerebellar flat maps: by doing so, some of the regions get a distorted importance (as shown in the mesh maps presented in the documentation). It would be good to provide a measure of the distortion to be expected.

Thank you for this point, it is indeed something we have considered at multiple stages. A representation that minimizes distortion is often more difficult to orient (e.g. by labeling axes). The choice to stick to a rectangle was thus made to improve readability and for aesthetic purposes. We have now additionally provided a map (in mm^2^) of the average distortion between folded and unfolded space in Figure 2 – Supplement 1.

11. Finally, while the software and documentation are very well organized, I was unable to run the app on the test data folders or my own data, using docker, singularity or the poetry installation option, which is absolutely required to complete this review. The 'getting started' section should also include a full processing script example on a test data set, outlining the main steps and basic parameters, especially as the toolbox is quite flexible and thus quite complex. Commands used for visualizing the various results should also be given in the 'Outputs' section, so users can visualize their data as in the examples. Given the richness of the manual delineation and segmentation effort, it would be valuable to release the training and testing data openly (note also that it is quite important for the U-Net step, where the training set properties have a strong impact on performance and potential biases).

We are surprised and disappointed the software did not work out-of-the-box, and we agree that it should be fully functional before publication. We have continued to improve the documentation as several collaborating labs have begun taking up this tool. Thus, we have found some common installation pitfalls and made improvements to the documentation accordingly, including a complete test-case example. We have now also provided additional pages on how to easily visualize these outputs in the online documentation.

We have now made the training dataset available online at https://zenodo.org/record/7007362.

12. In general, the paper is well written, but there are multiple areas that I have some issues with following the logical flow of what is being proposed. For example, the paper begins with demonstrating multiple metrics that are projected onto the hippocampal flatmap that includes thickness, myelin, curvature, gyrification, etc. It is unclear as to what information the authors want to convey here. This is the first mention of many of these multiple metrics as well and therefore their relevance is ultimately not extremely clear. As a result, it is hard to support their claim that "differences in morphological and quantitative features can be seen across the hippocampus, particularly across the subfields" as the goals of this particular figure are not at all clear.

Thank you for this comment, we have made some edits to try and improve the clarity and flow around Figure 2. It was our intention to show what we hope will become an increasingly standard view of the hippocampus, similar to how the neocortex is often represented in a semi-inflated surface view from the medial and lateral aspects. We felt that the differences seen in these quantitative measures is exciting as it aligns nicely to expected subfield borders with a clarity that is not often seen in hippocampal subfield imaging (for example some studies have observed subtle differences between predefined ROIs but it is rare to be able to differentiate, for example, CA1 from CA2 on the basis of thickness alone). We thus hoped to show off the types and clarity of information that could be gathered with HippUnfold.

13. Line 147: It is not totally accurate to state that ASHS makes use of multi-atlas registration as it also uses AdaBoost to correct for segmentation inaccuracies.

We have corrected this.

14. For the FreeSufer and ASHS comparisons – is it possible to provide some quantification of errors or anything like that? I think it would be helpful to quantify the differences in a more accurate manner. If this is in a previous publication and I missed it, it could be useful to reiterate here. The qualitative difference is nice – but there is room to compare them more quantitatively to one another.

As described in Response 3, we have now added more comparisons against Freesurfer and ASHS in a subset of 100 HCP Aging subjects, specifically examining subfield volume agreement, and trends with respect to aging (Figure 3). We also provide additional quantitative and qualitative visual comparisons for all subjects in Figure 3 – Supplement 1.

A more granular quantitative comparison, such as Dice scoresis somewhat challenging to do in detail since FS and ASHS both differ in the labels and the ways in which they simplify hippocampal subfields to suit the precision afforded by MRI. None of the three methods compared here can be validated against a ground truth subfield segmentation from the datasets examined here, which would require histological data. Thus, it's hard to say definitively what is an error or a simplification. Indeed, most MRI segmentation protocols rely on qualitative descriptions from histological work rather than quantitative information.

15. For the validation of the U-NET, details on the manual segmentation protocol, who did it, and its reliability are crucial. Training/testing paradigms would be helpful here. So would Bland-Altmann plots. I think in general the validation of these segmentations is quite poor – so more metrics that demonstrate the segmentation beyond dice overlaps would be helpful.

We have included additional rater details, as well as details about the protocol used. Validation of this protocol was performed in a separate paper (DeKraker et al., 2018), and we would like to note that the current paper is not a manual segmentation protocol but rather relies on this previously published work. Briefly, DeKraker et al. (2018) showed high intra- and inter-rater agreement, showed many qualitative features seen in histological reference materials, and included one same-subject MRI and histological validation from a temporal lobe epilepsy patient.

Training and testing protocols are described in the Materials and methods section “nnUNet training”, which has now been updated with additional details.

Bland-Altman plots comparing HippUnfold results to extant Freesurfer and ASHS methods are provided in Figure 3 and Figure 3 – Supplement 1.

With very high throughput segmentation, some manual errors are a certainty. By sticking to a fixed protocol we tried to avoid rater ‘drift’ towards a systematic bias, and in general we were pleased that UNet training appeared to generalize well across training samples. With successive iterations of training, we were able to identify and remove some common errors that may have stemmed from several low quality segmentations in the first training batch, but it is difficult to quantify this given the subjective nature of manual inspection and segmentation.

We have now made the training dataset available online at https://zenodo.org/record/7007362.

We encourage anybody who is interested to provide feedback or additional data that could be used as training data, which we see as an invaluable resource.

16. It is unclear how generalizable the method is outside of HCP acquisitions.

We included one non-HCP dataset (TSE-7T, n=70 L+R), on which automated QC showed good performance on nearly all subjects (see Automated error flagging sections). In addition, we have now included one temporal lobe epilepsy patient case.

We would also like to note that since the time of our initial submission, performance on other datasets and in different labs appears to be equally good. For example, see:

“Automated characterisation and lateralisation of hippocampal sclerosis from MRI in children with epilepsy” (first author Mathilde Ripart) at: Annual Meeting of the Organization of Human Brain Mapping, Glasgow, UK, 2022 [Editors' note: further revisions were suggested prior to acceptance, as described below.]

The manuscript has been improved but there are some remaining issues that need to be addressed, as outlined below:1. The description of some of the central methods used in the article (Laplacian embedding, shape injection) is too limited to understand fully how we obtain the unfolding. I see the point of 'increased complexity with increasing depth', but the article does not reach the level where the algorithms are explicitly described. It is also unclear if the authors used third-party software or their own implementation of these two methods.

We have decided to address this in two ways: we now include a new detailed Supplementary file detailing the algorithms used, AND we have added to the main text an intuitive explanation for how and why Laplace coordinates are used (“HippUnfold detailed pipeline” Materials and methods section). The latter was not requested by the Reviewer per se, but we have found it alleviates a lot of confusion about this issue.

“Intuition:

Imagine we have a tangled piece of wire. We attach one end to something hot (100OC) and the other to something cold (0OC), and then wait for the temperature to equilibrate along the entire wire. We then have a second wire that is tangled up in a different pattern (and possibly with a different length). We attach the same hot and cold endpoints, and wait for it to equilibrate as well. Then, when we want to find topologically homologous points between the two wires, we find the spot where they have the same temperature, say 10OC (or 10% its length), or the same for any other pair of homologous points. This explanation works since the heat equation describing the equilibrium temperatures is the same as the Laplace equation if we assume that the heat conductance (or thermal diffusivity) is constant. These wires make up a 1D example, but the same principle also applies to a folded 2D sheet, where the endpoints are edges rather than ends. Here we apply endpoints in two perpendicular directions: anterior-posterior (or HATA to ind.gris.) and proximal-distal (Sub to DG), making up a standardized 2D indexing system (or 'unfolded space').”

2. A second remaining issue I have is the somewhat puzzling fact that the T1w trained version of hippunfold performed better than the T2w one for the HCP-aging dataset: it would be good to understand why that is the case.

Please note that 1) T2w is actually more accurate with SRLM and GM segmentation (based on Dice shown in Figure 5), 2) the failure rate is higher for T2w but this is often related to image artifacts, failures in linear registration, or U-Net failures, 3) the context from surrounding tissue in the T1w images may be beneficial for avoiding U-Net failures.

We have added the following to the “Automated error flagging” Results section:

“It is interesting to note that fewer failures were observed in HippUnfold using T1w data compared to T2w data (Figure 7), even though performance of nnUNet tissue classification were slightly higher with T2w images (Figure 5) and these are more common in hippocampal subfield imaging literature. Examining some failed cases, we see that these often had poor image quality or artifacts, with subsequent issues like linear misregistration to the common CITI168 atlas or catastrophic nnUNet tissue classification failures.”

3. Finally, I still could not run the algorithm successfully on the provided example. Both Docker and Singularity provide very little information about why they fail. Installing and running the development version almost works, but only after installing several additional packages and non-standard research software from other groups (connectome workbench, NiftyReg, c3d…). None of these dependencies are described in the documentation. I would recommend the authors test their installation procedure on a bare-bones OS, for instance in a Linux virtual machine, as it is often challenging to remember what elements of a customized installation are being used or not. I am confident that the remaining issues are small, and that the usability of the software will be increased in the exercise.

The many different ways the app can be run were initially intended as features for flexibility, but we admit that it may also be a source of confusion. We have now tested HippUnfold on a fresh Linux (Singularity), Windows (Docker), and Linux Virtual Machine (Vagrant) environments. In all cases the pipeline works without software dependency issues since all required dependencies are in the HippUnfold container, but challenges may be faced when binding paths to the container, thus we have revised the “Getting Started” sections for each set-up with more specific details, including screencasts for the Virtual Machine option as well. Another issue often encountered is that pulling the singularity container may fail (since it can require a large amount of disk space and memory). As an alternative, we now also provide the latest release as a DropBox download. We have also added an additional “FAQ” section for issues that other collaborators have encountered in the past.

We are glad to hear the “pip install” worked, albeit with issues. The HippUnfold container (e.g. at Docker URI khanlab/hippunfold:v1.2.0) with all dependencies included, is the recommended method for end-users, but if the app is installed with pip, then the command-line run *must* include the `--use-singularity` flag, which will enable the workflow to download the dependency containers as needed for each rule in the workflow. The “Contributing to HippUnfold” documentation has now been updated to make this instruction more explicit. This setup executes the code in the same environment as the “pip installation” but with Snakemake handling these additional dependencies via Singularity. This setup is more complex, but works well when modifying the HippUnfold code while holding dependencies fixed, which is why we recommend standard users follow the Docker/Singularity instructions whereas advanced users wishing to further customize the code use “pip install”, or “poetry install”.